# The roles of vision and antennal mechanoreception in hawkmoth flight control

Ajinkya Dahake[1,2†], Anna L Stöckl[1†‡*], James J Foster[1], Sanjay P Sane[2], Almut Kelber[1]

[1]Vision Group, Lund University, Lund, Sweden; [2]National Centre for Biological Sciences, Tata Institute of Fundamental Research, Bangalore, India

**Abstract** Flying animals need continual sensory feedback about their body position and orientation for flight control. The visual system provides essential but slow feedback. In contrast, mechanosensory channels can provide feedback at much shorter timescales. How the contributions from these two senses are integrated remains an open question in most insect groups. In Diptera, fast mechanosensory feedback is provided by organs called halteres and is crucial for the control of rapid flight manoeuvres, while vision controls manoeuvres in lower temporal frequency bands. Here, we have investigated the visual-mechanosensory integration in the hawkmoth *Macroglossum stellatarum.* They represent a large group of insects that use Johnston's organs in their antennae to provide mechanosensory feedback on perturbations in body position. Our experiments show that antennal mechanosensory feedback specifically mediates fast flight manoeuvres, but not slow ones. Moreover, we did not observe compensatory interactions between antennal and visual feedback.
DOI: https://doi.org/10.7554/eLife.37606.001

*For correspondence:
anna.stoeckl@uni-wuerzburg.de

[†]These authors contributed equally to this work

Present address: [‡]Chair of Behavioral Physiology and Sociobiology, Würzburg University, Würzburg, Germany

Competing interests: The authors declare that no competing interests exist.

## Introduction

The impressive aerobatic manoeuvres of insects provide an insightful model for the neural control of flight (*Frye and Dickinson, 2001*; *Fuller et al., 2014*). Insect flight requires continual sensory feedback, both on the position of the body relative to the environment, as well as on perturbations to body position. Visual feedback provides key information about flight parameters including ground speed, distance to obstacles and targets, and aerial displacements (for a review see *Srinivasan et al., 1999*). However, visual estimation of self-motion (*Fuller et al., 2014*; *Hung et al., 2013*) is limited by its temporal resolution and substantial latency to flight muscle activation (*Sherman and Dickinson, 2004*; *Suver et al., 2016*). This may often be too slow to control very fast aerial manoeuvres, which require rapid sensory feedback before perturbations become uncontrollably large and thus energetically costly to the animals (*Bender and Dickinson, 2006*).

Avoiding the temporal limitations set by the visual system, insects use mechanosensors to sense their own motion, as these can transduce perturbations on much faster time scales (*Yarger and Fox, 2016*). The halteres of Dipteran insects are a classic example of gyroscopic function in active flight (*Fraenkel and Pringle, 1938*; *Nalbach, 1994*; *Pringle, 1948*). Halteres are club-shaped mechanosensory structures that were evolutionarily derived from the hind-wings. They vibrate at the wing beat frequency and can sense rotations in any axis, and provide crucial sensory input to stabilise flight after perturbations (*Ristroph et al., 2010*). Halteres, however, are a special feature of only Dipteran (and Strepsipteran) insects (*Pix et al., 1993*). How do flying insects from other orders, which also require fast feedback for stable flight, control flight manoeuvres without halteres? This question is especially interesting in insects active in dim light, as the visual systems of many insects trade off

temporal acuity for sensitivity (*Warrant, 1999*; *Warrant, 2017*), thus rendering visually based flight control even less reliable and increasing the need for mechanosensory feedback control.

Sphingids are a group of flying insects that are able to fly over a wide range of light intensities, due to their superposition compound eyes and additional neural adaptations (*O'Carroll et al., 1996*; *O'Carroll et al., 1997*; *Stöckl et al., 2017a*; *Theobald et al., 2010*). The effect of light intensity on their visual flight control has been quantified recently (*Sponberg et al., 2015*; *Stöckl et al., 2017a*). Moreover, the crepuscular hawkmoth *Manduca sexta* has been shown to use information provided by antennal mechanosensors, which may function similar to Dipteran halteres (*Sane et al., 2007*). The mechanosensory Johnston's organs, present at the pedicel-flagellar joint of the antennae, are stimulated by deflections of the antennal flagellum, and are sensitive to a wide range of frequencies (*Dieudonné et al., 2014*), which far exceed the temporal response range of the visual system (*Stöckl et al., 2017a*; *Theobald et al., 2010*). After ablation of their flagella, the Johnston's organs of *M. sexta* no longer receive relevant information, causing flight instability in these moths, whereas re-attachment of the flagella statistically significantly improves their flight performance (*Sane et al., 2007*). Impaired flight performance following flagellar ablation was also observed in other Lepidopteran species, such as the tortoise-shell butterfly *Aglais urticae* (*Gewecke and Niehaus, 1981*; *Niehaus, 1981*) and the diurnal swallowtail moth *Urania fulgens* (*Sane et al., 2010*). Although the above studies underscored the importance of antennal mechanosensors for natural flight, the severe behavioural impairment caused by flagellar ablation meant that the precise contributions of antennal mechanosensors to flight control remained an open question, as did their integration with the visual sense.

To address these questions, we chose an insect model which retains both the motivation and ability to fly after flagella ablation: the diurnal hawkmoth *Macroglossum stellatarum*. *M. stellatarum* feed from flowers while hovering in front of them, and previous studies have underscored the importance of visual feedback on their flower tracking behaviour (*Farina et al., 1995*; *Farina et al., 1994*; *Kern, 1998*; *Stöckl et al., 2017a*). Using this hawkmoth, we were able to test the role of antennal mechanosensors for the control of stationary hovering flight, as well as for flight manoeuvres at controlled temporal frequencies, focussing on the integration of visual and mechanosensory information (*Figure 1A*). Here, we show that in *M. stellatarum*, antennal mechanosensors play a key role in the control of hovering flight, specifically in the control of fast flight manoeuvres (rapid turns). Furthermore, we show that visual and antennal mechanosensory feedback operate in different frequency bands, with no sign of compensatory interaction.

## Results

### Antennectomised hawkmoths performed less stable hovering at a stationary flower

To quantify the role of antennal mechanosensory feedback in free flight, we trained hawkmoths (*Macroglossum stellatarum*) in a flight cage to approach and hover in front of an artificial flower with sugar solution provided in a nectary at its centre (see Materials and methods). Individual moths were tested in three antennal conditions: with intact antennae (*control*, blue, *Figure 1B* and *Figure 1—figure supplement 1*) with ablated flagella (*ablated*, red) and with re-attached flagella (*reattach*, green). The number of animals taking off and feeding from the flower decreased significantly with flagella ablation, but returned to control levels following reattachment (*Table 1*, *Supplementary file 1*). Yet, depending on light levels, a substantial proportion of individuals with ablated flagella still approached and fed from the flower (60% in bright and 36% in dim light), making it possible to study the combined roles of vision and antennal mechanosensory feedback on flight control in more detail.

When approaching the flower, moths with ablated flagella had distinctly longer and more tortuous flight trajectories than moths in the control and flagella re-attached conditions (*Figure 2—figure supplement 1*), pointing toward an impairment of flight control due to flagella ablation. To further quantify the effect of flagella ablation and re-attachement on flight performance, we initially focused on the hovering flight of the hawkmoth in front of a stationary flower, where its body position could be closely monitored (*Figure 1C*) and the target position was clearly defined by the position of the flower on which it fed. For these stationary flower experiment, we analysed the hovering flight of six

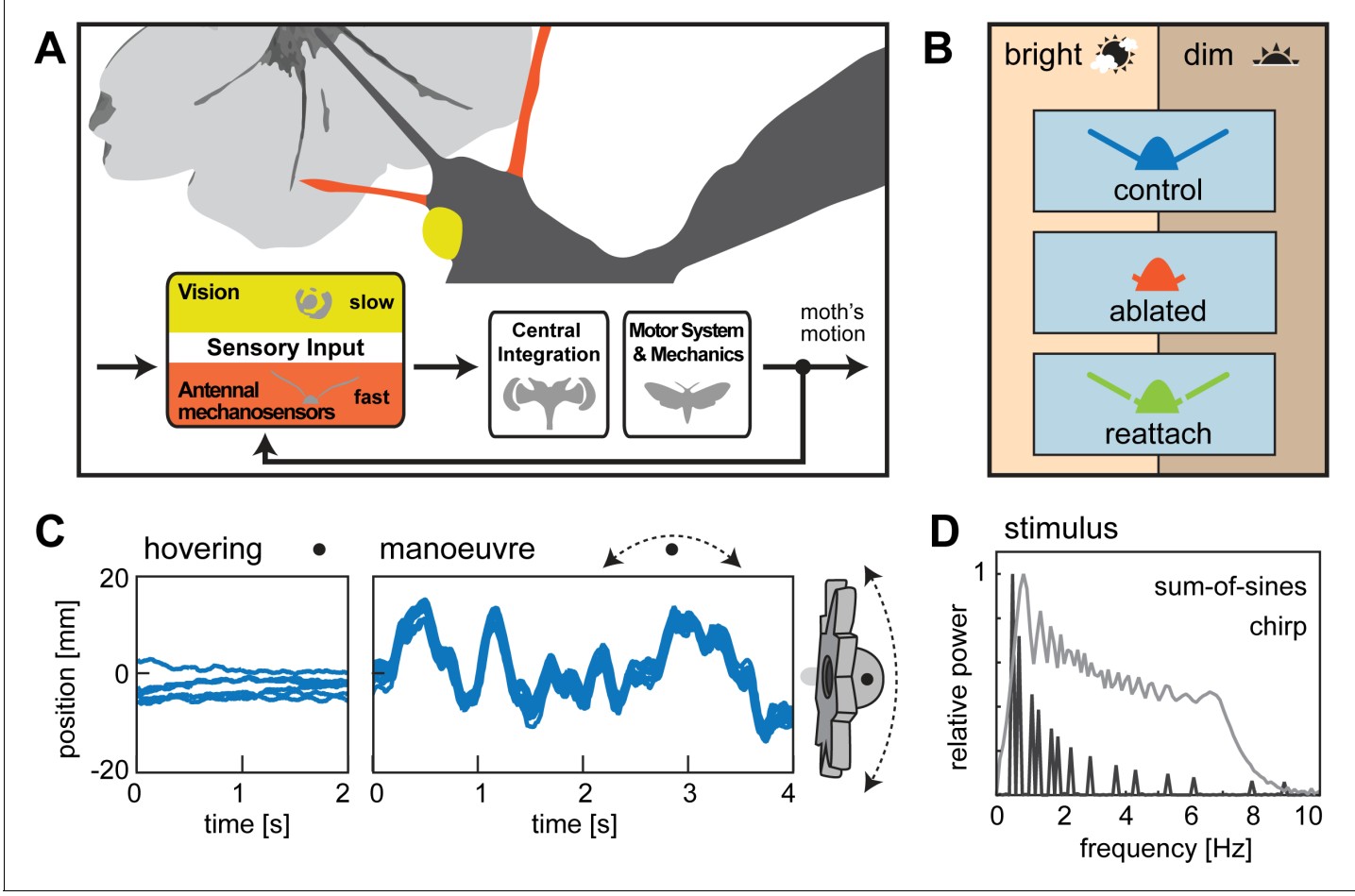

**Figure 1.** Flight control in hawkmoths requires vision and mechanosensation. (A) Flight control in insects requires sensory feedback on perturbations of body position. The visual system supplies such feedback, but with comparably long response latencies. In addition, insects use mechanosensory systems to control their position in the air, which provide rapid feedback and thus are crucial for fast flight manoeuvres. Here, we investigated the role of antennal mechanosensation and vision on flight control in the hummingbird hawkmoth *Macroglossum stellatarum*. (B) In order to quantify the effects of antennal mechanosensation in free flight, we subjected each hawkmoth to three treatments: intact antennae (*control*, blue), ablated flagella (*ablated*, red) and re-attached flagella (*reattach*, green). To quantify the role of vision, we tested these three antennal treatments in two different light intensities (*bright*: 3000 lux, corresponding to partially overcast daylight and *dim*: 30 lux, corresponding to sunset intensities). (C) All conditions were tested in free hovering flight at artificial flowers, which were either stationary (*hovering*) or moved at different temporal frequencies (*manœuvre*). (D) We used a stimulus composed of a *sum-of-sines* to sample distinct frequencies with similar velocities (amplitude adjusted accordingly), as well as a stimulus ramping up in frequency, while retaining similar amplitude (*chirp*).

DOI: https://doi.org/10.7554/eLife.37606.002

The following figure supplement is available for figure 1:

**Figure supplement 1.** Antennal surgery.

DOI: https://doi.org/10.7554/eLife.37606.003

animals, which performed in all three antennal conditions and in both light intensities (see Materials and Methods).

When hovering in front of the stationary flower, we noticed that ablated moths had a greater positional variation in relation to their target position than moths in the other two antennal conditions, as evident from their thorax position over time (blue line, *Figure 2A*). We quantified the amplitude of these thoracic movements across a range of frequencies from 0.5 to 50 Hz (Note that these frequencies are not related to flower movement, since the flower in this experiment remained stationary. The frequency analysis refers to the movement of the thorax of the animals). In all three antennal conditions, the amplitude of this thoracic 'jitter' decreased with increasing frequencies (i.e. the animals performed smaller movements at higher frequencies, *Figure 2B*; *Figure 2—figure*

**Table 1.** Proportion of hawkmoths performing specific behaviours across antennal conditions and light intensities.

Proportion of trials in which animals performed the following behaviours: *no flight*, *flight* (but no tracking of the flower), *tracking*. This dataset is based on the animals participating in the moving flower experiments. Of the total number of animals, 27 control, 22 flagella ablated, and 14 re-attached moths were tested in both light intensities. Some were tested multiple times to collect the necessary tracking data, and thus have contributed multiple trials to this dataset. Statistical comparisons were performed using multinomial regression including individual identity as a random factor, to model the rates of one of the three behaviours as a function of antennal condition and light intensity (without interaction terms). Statistical significance is indicated by: *p < 0.05, **p < 0.01, ***p < 0.001. For statistical details, see *Supplementary file 1*.

| | | Control | Ablated | Reattach |
|---|---|---|---|---|
| bright | No flight | 0.03 | 0.11 | 0.05 |
| | Flight | 0.15 | 0.29 | 0.1 |
| | Tracking | 0.82 | 0.60 *** | 0.85 |
| | Total | 38 | 42 | 20 |
| dim *** | No flight | 0.03 | 0.34 | 0.08 |
| | Flight | 0.09 | 0.30 | 0.08 |
| | Tracking | 0.88 | 0.36 *** | 0.84 |
| | Total | 35 | 66 | 25 |

DOI: https://doi.org/10.7554/eLife.37606.004

supplement 2A). At 3000 lux, the thoracic jitter of ablated moths was statistically significantly larger than that of the other two antennal conditions between 0.7 and 5 Hz, and between 8 and 11 Hz (*Figure 2B*, *Supplementary file 2*), whereas the difference between control and re-attached condition was not statistically significant. Thus, re-attaching the flagellum restored flight performance close to the control state.

Because many insects, including hawkmoths, use abdominal movements for aerial stabilization during flight (*Camhi, 1970*; *Dyhr et al., 2013*; *Hinterwirth et al., 2012*), we also quantified the movement of the abdomen over the same frequency range. Across all three antennal conditions, the abdominal and thoracic movements revealed similar trends; in flagella ablated moths, their magnitude was statistically significantly larger than in control moths or flagella re-attached moths over the entire range of frequencies tested (*Figure 2C*, *Supplementary file 3*). Like thoracic jitter, abdominal jitter of moths with re-attached flagella was not statistically significantly different from control moths (except for one frequency: 1.66 Hz).

Since hovering is a dynamically unstable flight mode (*Liang and Sun, 2013*; *Wu and Sun, 2012*), hovering animals need continual sensory feedback to maintain a fixed position (*Cowan et al., 2014*). Both the visual system and antennal mechanosensory systems could provide sensory feedback to correct for deviations from the target position. Because flagella ablated moths showed larger positional jitter, especially at higher frequencies, we conclude that antennal mechanosensory feedback is required for the control of hovering flight. Without antennal input, the feedback about deviations from the target position, likely supplied by the visual system and therefore slower, causes moths to drift further from their target position before a corrective manoeuvre can be initiated. This in turn results in greater thoracic and abdominal movements.

## Flagella ablation reduced flower tracking performance at high flower movement frequencies

After observing impaired flight stability during flower approach and stationary hovering with flagella ablation, we went on to examine its effects on flight manoeuvres at specific temporal frequencies. To this aim, we moved the artificial flower along a controlled trajectory while the hawkmoths were feeding from the nectary, thus eliciting flight manoeuvres of controlled frequencies and amplitudes while the moths were tracking the flower (*Figure 1C*). To probe the moths' manoeuvrability at

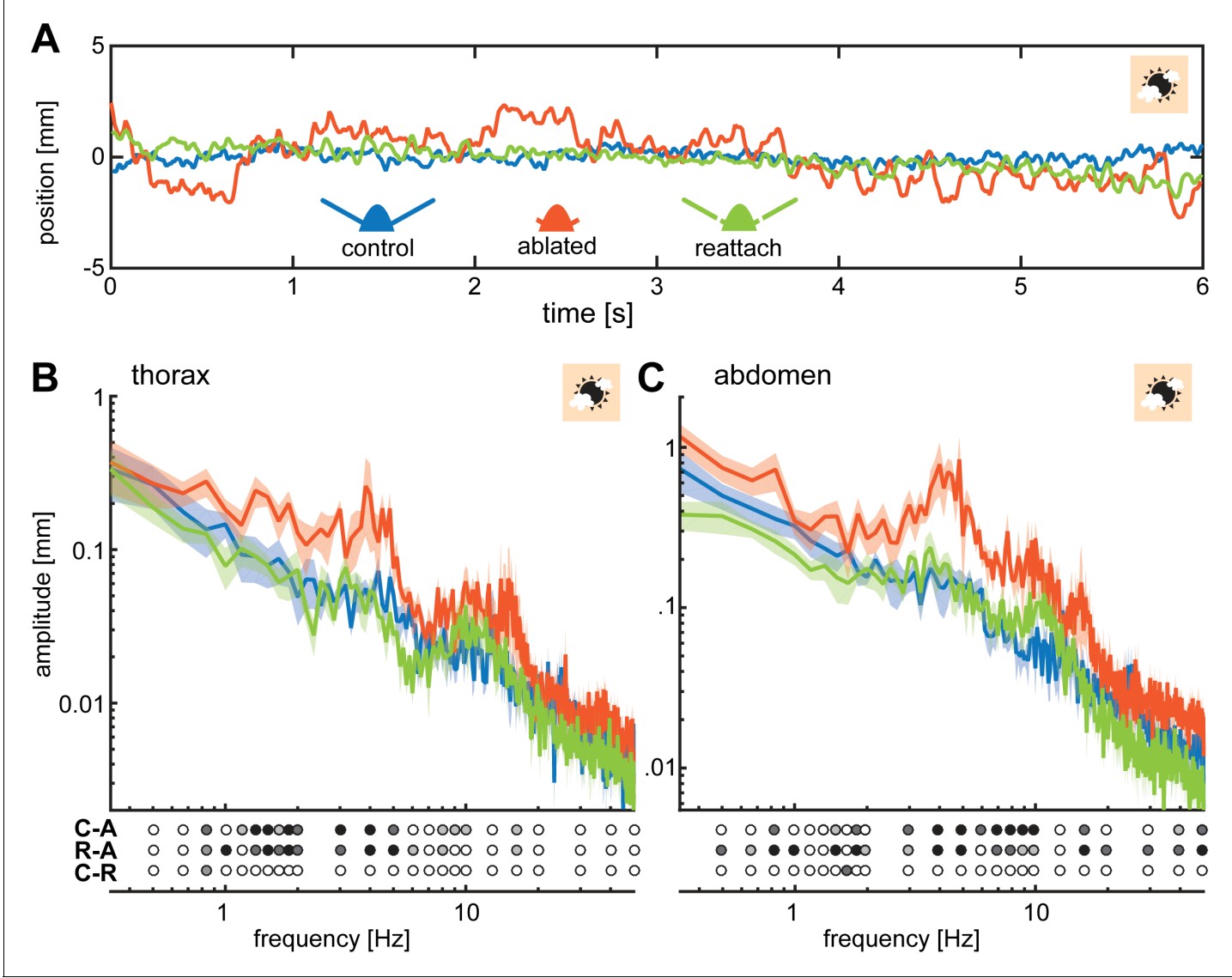

**Figure 2.** Flagella ablated hawkmoths showed greater thorax and abdomen movements during hovering flight at a stationary flower. (A) When hawkmoths hovered in front of a stationary flower at 3000 lux, it was notable that flagella ablated moths jittered around their target position with larger amplitudes than moths of the other two antennal conditions, as quantified by the position of their thorax. The nectary is centered at 0 mm in this graph. (B) The thorax of moths with ablated flagella jittered with significantly higher amplitudes than the other two antennal conditions at frequencies between 1 and 5 Hz. There was no significant difference between control and re-attached moths. (C) The position of the abdomen in the three antennal treatments showed a similar trend to the thorax: the flagella ablated moths exhibited significantly larger abdomen jitter in the frequency range between 0.5 and 10 Hz than the other two treatments. (B, C) Lines show average, and shaded areas ± SEM. Statistical significance is indicated below the plots as: black p < 0.001, dark grey: p < 0.01, grey p < 0.05, white p > 0.05. Post-hoc tests were performed as part of a general linear model including antennal treatment and frequency (binned to the logarithmic scale) as factors, see **Supplementary file 2** and **Supplementary file 3**.
DOI: https://doi.org/10.7554/eLife.37606.005

The following source data and figure supplements are available for figure 2:

**Source data 1.** Contains the source data for the frequency spectra shown in **Figure 2** and **Figure 2—figure supplement 2**.
DOI: https://doi.org/10.7554/eLife.37606.008
**Figure supplement 1.** Hawkmoth flower approach.
DOI: https://doi.org/10.7554/eLife.37606.006
**Figure supplement 2.** Flagella ablated hawkmoths performed less stable hovering at a stationary flower at 30 lux.
DOI: https://doi.org/10.7554/eLife.37606.007

different amplitudes and speeds of flower movement, we used two movement patterns: one pattern was generated from a sum of sine-waves ranging from 0.5 to 8.9 Hz. They decreased in amplitude with increasing frequency to retain a constant velocity ('sum-of-sines', *Figure 3A*), which allowed the hawkmoths to track the entire stimulus successfully. The second movement pattern had a constant amplitude, while its frequency increased over time from 0 to 7.3 Hz over time ('chirp', *Figure 1D and 3C*), thus resulting in increasing flower velocity. This stimulus was designed to test the limits of the hawkmoths' manoeuvrability, as the increasing velocity made it more challenging for them to track the artificial flower. We analysed the flight performance of 12 moths, which tracked both stimuli in all antennal conditions and two light intensities.

When tracking the sum-of-sines stimulus, tracking performance of control hawkmoths was consistent with previous investigations of intact individuals of this species (*Farina et al., 1995*; *Farina et al., 1994*; *Stöckl et al., 2017a*). To quantify the accuracy of the tracking performance, we used a metric that evaluates their accuracy in tracking both the amplitude (*Figure 3—figure supplement 1D–F*) and the phase (*Figure 3—figure supplement 1G–I*) of the flower movement, termed tracking error (*Roth et al., 2014*; *Sponberg et al., 2015*). In all antennal conditions, the control moths tracked the sum-of-sines stimulus accurately at low flower frequencies (*Figure 3A*): at 3000 lux, their tracking errors were close to 0 for flower movements up to 1 Hz, indicating nearly perfect tracking (*Figure 3B*). With increasing frequency, tracking errors increased, but there was no statistically significant difference in tracking error between antennal conditions for frequencies below 2 Hz (*Figure 3B*, *Supplementary file 4*). At higher flower frequencies, flagella ablated moths overshot the flower movements, resulting in a greater lag between the position of the moth and the flower (*Figure 3—figure supplements 1,2*) and thus larger tracking errors (*Figure 3B*): in the range of 2 to 5 Hz, tracking errors were statistically significantly higher for the flagella ablated moths than for both the control moths and moths with re-attached flagella (*Supplementary file 4*). In this flower frequency range, hawkmoths with re-attached flagella also had statistically significantly higher tracking errors than control moths. Thus, the reduction of antennal mechanosensory feedback impaired flight control specifically at the higher temporal frequencies of flower movement, which compelled the moths to perform faster turns. The ability of flagella ablated moths to track at frequencies below 2 Hz suggests that vision (and possibly other sensory modalities) provide feedback that is sufficiently fast to enable control of slower manoeuvres.

To ensure that the differences in flight performance between the three antennal conditions was independent of the specific type of flower movement, we presented the same hawkmoths with a 'chirp' stimulus in which the amplitude of flower movement was held constant, while the temporal frequency continuously increased from 0 to 7.3 Hz (*Figure 1D*), and with it the velocity of the flower. Unlike the sum-of-sines stimulus, which moths in all antennal conditions were able to track in its entirety, this stimulus was designed to test the limits of the hawkmoths' manoeuvrability, as the increasing velocity made it increasingly difficult for the moths to track the flower. At low flower frequencies, hawkmoths of all antennal conditions tracked this stimulus with high fidelity (*Figure 3C*). However, as the flower frequency increased, flagella ablated moths tended to overshoot the position of the flower at the end of each sideways movement, when the flower movement changed direction. The accumulated phase lag and overshoot were eventually large enough to cause the moths to lose contact with the nectary and abort flower tracking (*Figure 3C*). Only 1 out of 12 ablated moths succeeded in following the flower movement during the entire stimulus at 3000 lux. In contrast, all control and 10 out of 12 re-attached moths tracked the flower until the maximum frequency. As a measure of effective flight performance, we quantified the flower frequency at which hawkmoths in the different antennal conditions aborted flower tracking (*Figure 3D*). At 3000 lx, there was no statistically significant difference between the control and re-attached flagella moths (*Supplementary file 5*). Flagella ablated hawkmoths aborted flower tracking at a median frequency of only 4.4 Hz, statistically significantly lower than the other two conditions (*Figure 3D*, *Supplementary file 5*). While re-attached hawkmoths did not show a statistically significant difference in tracking abortion frequency from control moths, looking at the flight tracks tracking the chirp stimulus in more detail (*Figure 3—figure supplement 3*) revealed some differences in tracking performance: the power they shared with the stimulus at frequencies above 4 Hz was higher than that of ablated moths, but lower than that of control moths (*Figure 3—figure supplement 3E*), and similarly, their phase delay did not increase as quickly with frequency as that of ablated moths, but quicker than for control moths (*Figure 3—figure supplement 3E*).

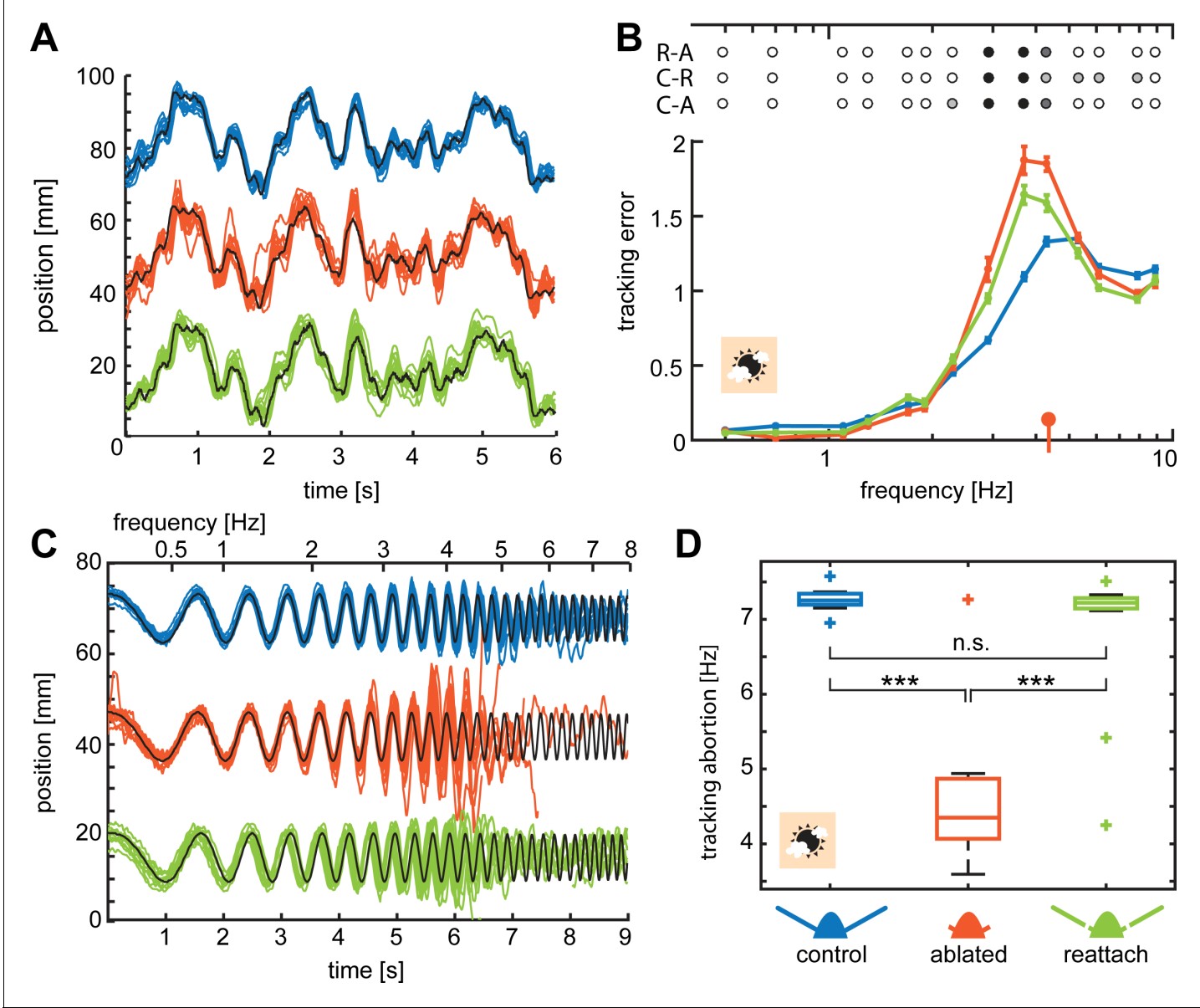

**Figure 3.** Flagella ablated hawkmoths showed reduced tracking performance of flowers moving at high frequencies. (A, C) Trajectories of hawkmoths tracking moving flowers with the sum-of-sines (A) and the chirp (C) stimulus. Trajectories of the different antennal conditions are stacked for comparability. When tracking a moving robotic flower at 3000 lux, hawkmoths with ablated flagella often overshot the movements of the flower, specifically at higher frequencies. With increasing frequencies, moths also increasingly lagged behind the phase of flower movements more strongly. While the amplitude in the sum-of-sines stimulus was adjusted such that moths of all conditions could track the entirety of the stimulus, the chirp stimulus forced moths with too large overshoots and phase-lags to loose contact with the flower, and abort tracking (see red tracks in C). (B) Together, overshooting and phase-lags resulted in an increased tracking error of flagella ablated moths with the sum-of-sines stimulus at frequencies between 2 and 6 Hz, compared to both the control and re-attached condition. Linear mixed-effects models were used to compare the tracking error of the different antennal treatments with respect to frequency. Colours indicate significance (black $p < 0.001$, dark grey: $p < 0.01$, grey $p < 0.05$, white $p > 0.05$, **Supplementary file 4**). The red indicator on the x-axis gives the median frequency at which flagella ablated moths aborted tracking the chirp-stimulus (D). Curves show the mean and 95% confidence intervals of the mean, calculated in the complex plane. (D) For the chirp stimulus, we compared the movement frequency of the flower, at which the moths aborted tracking across antennal treatments, showing that flagella ablated moths lost contact with the flower at significantly lower frequencies than the control and re-attached condition. A Friedman test was used to compare between the treatments (***$p<0.001$, **: $p<0.01$, *$p<0.05$, **Supplementary file 5**).

DOI: https://doi.org/10.7554/eLife.37606.009

The following source data and figure supplements are available for figure 3:

*Figure 3 continued*

**Source data 1.** Contains the source data for the original traces of flower and moth for all moving flower experiments shown in *Figure 3*, and further analyzed in *Figure 4A,B* and *Figure 3—figure supplements 1,3,4*.

DOI: https://doi.org/10.7554/eLife.37606.014

**Source data 2.** Contains the source data complex valued responses of moths tracking the sum-of-sines stimulus shown in *Figure 3* and *Figure 3—figure supplement 1*.

DOI: https://doi.org/10.7554/eLife.37606.015

**Figure supplement 1.** Hawkmoth tracking performance with the sum-of-sines stimulus in bright and dim light.

DOI: https://doi.org/10.7554/eLife.37606.010

**Figure supplement 2.** Frequency analysis of sum-of-sines stimulus tracking performance.

DOI: https://doi.org/10.7554/eLife.37606.013

**Figure supplement 3.** Frequency analysis of chirp stimulus tracking performance.

DOI: https://doi.org/10.7554/eLife.37606.012

**Figure supplement 4.** Flower tracking performance with chirp stimuli in dim light.

DOI: https://doi.org/10.7554/eLife.37606.011

Despite the differences in the movement patterns of the two stimuli and their demands on flower tracking, we observed similar trends in hawkmoth flight performance. Moths in all antennal conditions tracked flower movements well at low frequencies, whereas flagella ablated hawkmoths were statistically significantly impaired at higher frequencies compared to the other two antennal conditions. The average flower frequency at which ablated moths failed to track the chirp stimulus was consistent with the frequency range for which the tracking error with the sum-of-sines-stimulus was greatest, despite the difference in flower velocity between stimuli. These data show that antennal feedback is crucial for fast turns - or directional changes - which are associated with changes in body posture.

## Slow visual feedback impaired flower tracking performance of all antennal conditions

In the experiments described so far, hawkmoths of all antennal conditions did not differ in their flight performance when performing slower movements (at lower frequencies of positional jitter when hovering at a stationary flower, and at flower frequencies when tracking a moving flower). At these movement frequencies, feedback from other sensory modalities likely mitigated the problems in flight control caused by antennal ablation. In particular, visual feedback is known to provide information about changes in insect body (head) position in flight (for a review see *Srinivasan et al., 1999*), albeit with longer latencies and a lower frequency range than mechanosensory feedback (*Sane et al., 2007*). Because the latency of visual feedback depends on the ambient light intensity (*Stöckl et al., 2017a*), we next tested how the reliability of visual feedback affected the hawkmoth's flight performance by decreasing the ambient light intensity in combination with the antennal manipulations. We tested the same group of hawkmoths at an illumination of 30 lux, close to the light intensity limit at which these diurnal hawkmoths are still able to reliably approach and feed from the artificial flowers (*Stöckl et al., 2017a*). If flagella ablated hawkmoths relied mainly on visual feedback for flight control when they lack antennal mechanosensory feedback, their flight performance should be poorer under low light, as compared to bright light. However, we did not expect any difference in the performance of hawkmoths in the control and re-attached conditions, because these moths receive fast feedback from their antennal mechanosensors.

To quantify the effect of light intensity on flight performance during stationary hovering, we calculated the difference in the amplitude of thoracic and abdominal movements for individual moths of all three antennal conditions (*Figure 4C,D*). We found no statistically significant effect of antennal condition on the average difference in thorax (*Supplementary file 7*) or abdomen (*Supplementary file 13*) jitter between dim and bright light, suggesting that differences in the temporal acuity of visual inputs did not additionally affect the impact of flagella ablation on flight performance.

We then went on to quantify the effect of light intensity on flight performance in the moving flower experiments. While we did observe an increase in tracking error of flagella ablated hawkmoths with the sum-of-sines stimulus in dim light (*Figure 3—figure supplement 1B*), particularly at

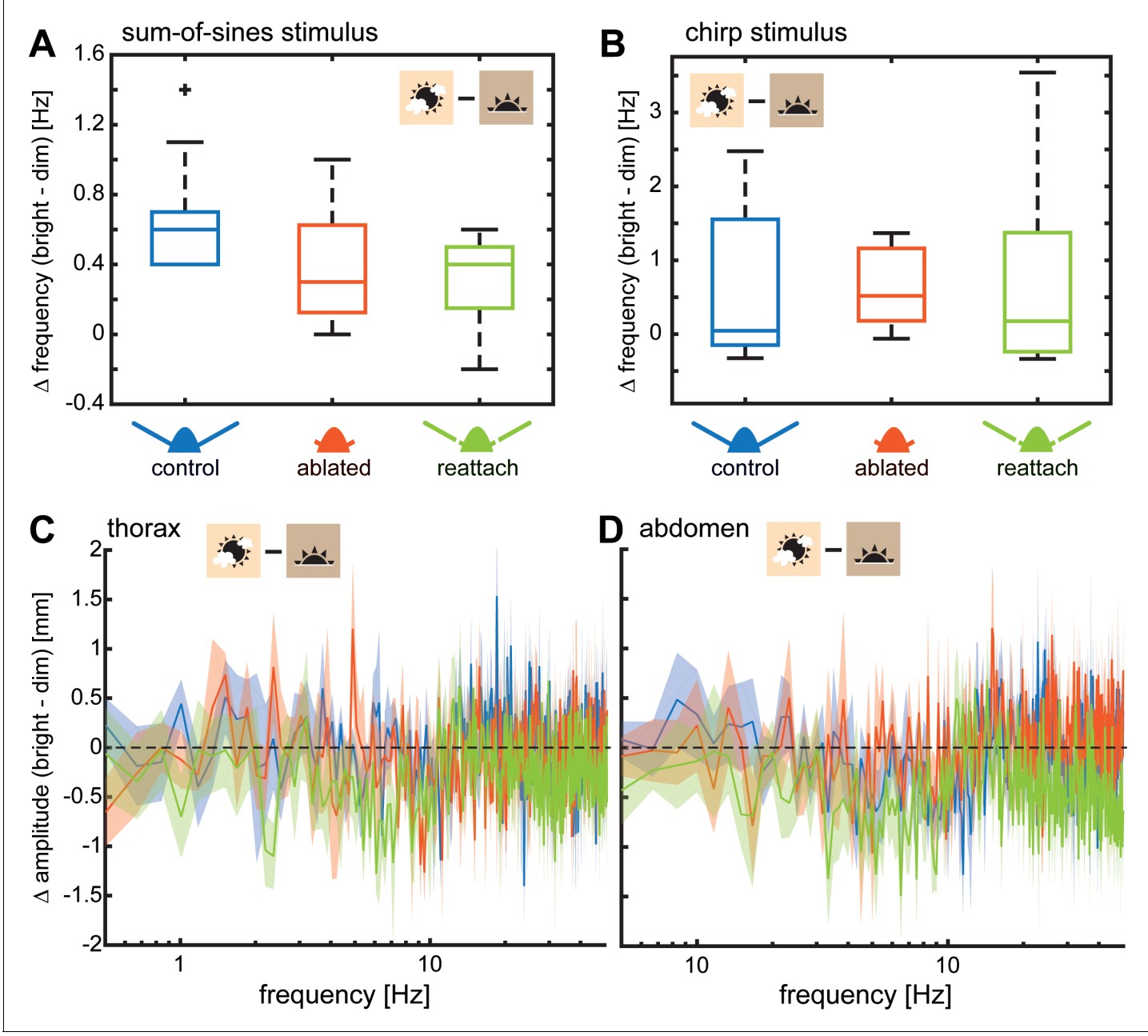

**Figure 4.** Light intensity had the same effect on all antennal treatments. To test the effect of visual feedback and its possible interaction with antennal mechanosensory feedback on flower tracking, we performed all experiments both in bright (3000 lux) and dim (30 lux) light intensities. Hawkmoths showed reduced tracking performance of artificial flowers moving at higher frequencies in dim light, due to the slowing of their visual system (*Figure 2—figure supplement 2*, *Figure 3—figure supplements 1,4*). Here, we compare tracking performance between bright and dim light across antennal treatments. (A) We quantified the difference in frequency between light intensities at which moths reached a tracking error of 1 with the sum-of-sines stimulus. There was no significant difference (*Supplementary file 8*) between antennal conditions, suggesting that vision reduced tracking performance in dim light irrespective of the presence or absence of mechano-sensory feedback. (B) Similarly, there was no significant difference between the tracking performance in dim and bright light for the chirp stimulus (quantified as the difference of tracking abortion frequency at the two light conditions) (*Supplementary file 6*). (C–D) We determined the difference between the log-transformed magnitude spectra for thorax (C) and abdomen (D) jitter in bright and dim light. No significant effect of antennal condition was found using Friedman comparisons of the average difference in thorax or abdomen movements (*Supplementary files 7* and *13*). Lines show average, and shaded areas ± SEM.

DOI: https://doi.org/10.7554/eLife.37606.016

lower flower frequencies, we observed the same effect in the control and flagella re-attached conditions (*Figure 3—figure supplement 1A,C*). We quantified the effect by comparing the difference in flower frequency at which each individual reached a tracking error of 1 in dim and bright light across antennal conditions (*Figure 4A*). There was no significant difference between antennal conditions (*Supplementary file 8*). Thus, light intensity affected flower tracking in general (as has been shown previously, (*Sponberg et al., 2015*; *Stöckl et al., 2017a*), but did not interact with antennal condition. Experiments using the chirp stimulus further confirmed this finding (compare *Figure 3D* and *Figure 3—figure supplement 4*). To compare the performance of moths in dim and bright light, we measured the difference in flower frequency at which moths aborted tracking with the chirp stimulus in dim and bright light across antennal conditions (*Figure 4B*). Also for this stimulus, we did not find statistically significant differences between antennal conditions (*Supplementary file 6*), indicating that visual feedback did not compensate for the loss of mechanosensory feedback in flagella ablated moths. Instead, the slower visual processing affected flight control similarly in all antennal conditions.

## Discussion

Although visual and mechanosensory feedback is known to play a prominent role in the control of insect flight, it is not clear how these inputs are integrated by the insect brain to generate behaviour. In Dipteran flies, which use halteres as gyroscopic sensors, vision and mechanosensation operate in frequency ranges that are complementary (*Mureli and Fox, 2015*; *Yarger and Fox, 2016*). A natural question arising from these studies is: how do insects that lack halteres process mechanosensory and visual feedback? To address this question, we here investigated how visual inputs from compound eyes and mechanosensory inputs from antennal Johnston's organs control flight in combination. For both stationary hovering and flight manoeuvres during flower tracking in *Macroglossum stellatarum*, our data show that antennal mechanosensory input is crucial for control of fast flight manoeuvres, while visual input controls the slower ones - similar to observation in flies.

### Flagellar re-attachment improves flight performance

We have shown that flight control in the diurnal hawkmoth *M. stellatarum* requires feedback from antennal mechanosensors. As also observed in previous experiments by *Sane et al. (2007)*, re-attaching the flagellum restored flight performance by reloading the Johnston's organs, both for stationary hovering and flower-tracking behaviors. This is consistent with the growing body of evidence (*Dieudonné et al., 2014*; *Gewecke and Niehaus, 1981*; *Niehaus, 1981*; *Sane et al., 2010*) that Lepidoptera use antennal Johnston's organs for flight control. One possible way how antennal mechanosensors might impact flight control is by providing feedback for head stabilisation. While Hymenopteran insects (*Polistes humilis*) seem to purely rely on visual information to stabilise their head during roll manoeuvres (*Viollet and Zeil, 2013*), preliminary data from hawkmoths shows that they might require antennal feedback for head stabilisation (*Sane et al., 2018*).

As demonstrated previously (*Sane et al., 2007*), while flagellum re-attachment improved flight performance statistically significantly compared to the flagella ablated condition, it did not restore it to the level of intact animals in all experimental conditions. Feedback provided with re-attached flagella restored flight performance when hovering at stationary flowers to levels that were statistically not significantly different from the control group (*Figure 2*). Similarly, moths with re-attached flagella tracked the chirp stimulus at the moving flower for similar lengths as control animals (*Figure 3C,D*) – although impairments in flower tracking at higher flower movement frequencies compared to the control condition were visible upon a more detailed analysis of the flight tracks (*Figure 3—figure supplement 3*). There were also statistically significant differences in the flight performance of re-attached and control moths with the sum-of-sines stimulus (*Figure 3A,B*). These differences between experimental conditions suggest that flagella re-attachement did not entirely restore flight performance back to the levels of intact animals. One reason for this might be that some properties of the re-attached flagella differed from those of intact animals. Re-attached flagella were not connected to the haemolymph system of the hawkmoth and thus dried out, which reduced their weight by more than 50% (see Materials and methods). Since the flagella are thought to provide a mass that inertial forces act on (*Sane et al., 2007*), changes in weight may considerably alter the sensory input to the Johnston's organs. Changes in the flexibility of the flagella due to moisture loss may also

contribute to this effect. Moreover, there are also mechanosensors along the length of the flagellum, which may be important for flight control. When the antennal nerve is severed, these mechanosensory units remain inactive even after flagellar reattachment, which may add to the observed deterioration in their ability to control flight. The roles of these mechanosensors and of the weight and flexibility of the flagella need to be further explored in future experiments.

## Antennal mechanosensation and vision operate in different frequency bands

Our experiments quantified the frequency range in which antennal mechanosensory feedback is required for the control of flight in *M. stellatarum* moths using a moving flower which the animals tracked to initiate flight manoeuvres at different temporal frequencies. We demonstrated that flagella ablated hawkmoths can track flowers moving at frequencies below 2 Hz with the same fidelity as hawkmoths with intact antennae (*Figure 2 and 3*). This suggests that control of slower manoeuvres is not as dependent on antennal mechanosensory feedback, as is the control of faster manoeuvres. On the other hand, flagella ablated hawkmoths performed statistically significantly worse than moths with intact and re-attached flagella at flower movements above 2 Hz, where more rapid turns are required to follow the lateral trajectory of the moving flower. Our findings are mirrored in the study of Dipteran flight control: slower rotations of fruit flies are tuned stronger to visual feedback, whereas faster rotations require feedback from haltere mechanosensors (*Sherman and Dickinson, 2003*).

It is noteworthy that the response delays in the flower tracking of *M. stellatarum* we observed were very short compared to visuomotor feedback loops measured in other hawkmoths (*Sponberg et al., 2015*) and other flying insects (*Reiser et al., 2012*; *Viollet and Zeil, 2013*): with the sum-of-sines stimulus at 3000 lux, hawkmoths in the control condition showed a phase lag of approximately 90° at the highest temporal frequency of 8.9 Hz (*Figure 3—figure supplements 1,2*), suggesting a response delay of less than 30 ms. Similarly, the phase delay with the chirp stimulus at 6.5 Hz in the control condition at 3000 lux was close to 80° (*Figure 3—figure supplement 3*), indicating a response delay of 35 ms. This would indeed be one of the fastest visuomotor transformations described in insects. Considering that visually tracking the flower likely involves computing a directional motion component, and the fastest latencies of wide-field motion neurons in blowflies are approximately 25 ms (*Warzecha and Egelhaaf, 2000*), it is unlikely that such fast flower tracking responses are purely elicited by vision in *M. stellatarum*. Recent work by *Roth et al. (2016)* in *Manduca sexta* has demonstrated that mechanoreceptors on the proboscis can play a role in monitoring flower position, in addition to visual input. Mechanosensors have much faster transduction than most visual receptors. We cannot exclude a mechansensory component to flower tracking originating from the proboscis in *M. stellatarum*, but the phase lags observed in *M. sexta* for purely mechanosensory tracking, even though shorter than those for purely visual tracking at higher temporal frequencies, are still distinctly larger than the ones observed for flower tracking in our experiments (*Roth et al., 2016*). Another possible explanation for the short tracking delays at high frequencies is a direct mechanical coupling between the head of the hawkmoth and the flower via the proboscis. The flowertracking responses of the hawkmoths might include this mechanical coupling, which could explain their extraordinarily fast responses at high flower movement frequencies. The change in responses across antennal conditions and light intensities shows that there is, nevertheless, a strong sensorimotor component of the behaviour, and since the potential mechanical coupling, as well as putative mechanosensory input from the proboscis, were present in all antennal conditions as well as light intensities, they did not affect the observed results with respect to visual and antennal mechanosensory feedback. Moreover, we did observe changes in flight performance upon antennal ablation and re-attachement both during flower approach (*Figure 2—figure supplement 1*) and hovering at stationary flowers (*Figure 2*, *Figure 4C,D*), where mechanosensory inputs and mechanical coupling did not play a role.

Thus, we conclude that mechanosensory feedback from the antennae is essential for the control of fast flight manoeuvres, which require corrective movements to occur in timescales that may not be sufficient for the transduction of visual feedback. This again is analogous to the finding that the control of fast saccadic rotations in Dipterans mainly requires mechanosensory feedback from the halteres, while vision plays a relatively marginal role (*Bender and Dickinson, 2006*; *Sherman and Dickinson, 2003*).

# Vision does not compensate for the loss of antennal mechanosensation in hawkmoth flight control

Both vision and mechanosensation contribute to insect flight control, and the mechanistic underpinnings of this multimodal integration are subject of many ongoing investigations. In Dipteran flies, vision and haltere mechanosensation operate in complementary frequency ranges, and while both inputs are required for stable flight under most circumstances (*Yarger and Fox, 2016*), they do not seem to compensate for each other (*Mureli and Fox, 2015*). Antennal movements also depend on feedback from multiple sensory modalities. For example, in honeybees, airflow on the antennae and optic flow influence antennal positioning in tethered as well as free flight (*Roy Khurana and Sane, 2016*). In the Oleander hawkmoth *Daphnis nerii*, visual feedback modulates antennal positioning in a similar way (*Krishnan and Sane, 2014*).

Here, we tested how vision and antennal mechanosensation in combination influence flight control during flower tracking. Using a bright and a low light intensity, we manipulated the temporal resolution of visual responses (*Stöckl et al., 2017a*; *Stöckl et al., 2016*). In dim light, the low speed and reduced reliability of the visual input to flight control causes larger tracking errors when flowers move at high frequencies for all antennal conditions (*Figure 4*). This effect is explained by the fact that visual input is essential for moths to identify and track the flower movement relative to their own position – antennal mechanosensors cannot provide the required information (*Figure 1A*). Because visual processing is slower in dim light, moths face greater difficulties in resolving fast flower movements, which causes failure in tracking (*Sponberg et al., 2015*; *Stöckl et al., 2017a*).

We did not observe a specific effect of light intensity on flight control in the flagella ablated moths. This suggests that, even at higher resolution under brightly lit conditions, visual feedback is unable to mitigate the instability caused by the loss of antennal mechanosensory feedback. Two main hypotheses could explain this finding: first, the contributions of vision and mechanosensation contribute to the motor outputs via separate parallel pathways, whose functions do not overlap. This is unlikely, as recent recordings of descending neurons in Oleander hawkmoth show that they respond to both visual and mechanosensory stimulation (*Mohan et al., 2017*). Alternatively, vision and mechanosensation share descending pathways but operate in different frequency ranges, and the visual input is too slow to compensate for the lack of antennal mechanosensory feedback. The latter hypothesis is consistent with physiological studies showing that mechanosensors in Johnston's organ respond to antennal displacements at frequencies of up to 100 Hz in the hawkmoth *M. sexta* (*Sane et al., 2007*), whereas the wide-field motion-sensitive neurons of the same species cease to respond at temporal frequencies above 20 Hz (*Stöckl et al., 2017a*), at which most mechanosensors of the Johnston's organ only show a weak response. Eventually, an assessment of the physiological responses of descending neurons that activate the flight muscles is required to reveal the mechanisms of integration of visual and mechanosensory information in control of flight in hawkmoths.

## Conclusion

Antennal mechanosensation represents one strategy for flying insects to obtain rapid sensory feedback about changes in self-motion, which is crucial for flight control. We showed here that in the diurnal hawkmoth *M. stellatarum*, mechanosesory feedback from antennae is required for the control of fast flight manoeuvres and rapid deviations from their hovering position, whereas their visual system drives the control of slower manoeuvres. These findings detail a striking similarity to the interaction between mechanosensory halteres and vision in the Dipteran flight control model, and for the first time dissect the combined role of visual and antennal mechanosensory feeback for flight control in hawkmoths, which may be representative for many other non-Dipteran insects.

## Materials and methods

### Animals

Wild adult *Macroglossum stellatarum* L. (Sphingidae), were caught in Sorède, France. Eggs were collected and the caterpillars raised on their native host plant *Gallium sp.* The eclosed adults were allowed to fly and feed from artificial flowers similar to the experimental flowers, in flight cages (70 cm length, 60 cm width, 50 cm height) in a 14:10 hr light:dark cycle for at least one day before experiments.

All animals were tested with intact antennae first (*control*), then with ablated flagella (*ablated*), and finally with re-attached flagella (*reattach*) as described below (*Figure 1—figure supplement 1*). Only data from animals that could be tested under all three antennal conditions was included in the final data analysis.

## Surgery: flagella ablation and re-attachment

For flagella ablation, moths were held, by their thorax under a dissection microscope and their flagella were clipped with a pair of surgical scissors, while retaining 5–10 annuli (*Figure 1—figure supplement 1B*). This ensured that Johnston's organs, located at the base of the antennae, were left intact but unloaded. Ablated flagella were preserved in a plastic petri dish with wet tissue to prevent them from drying and losing shape until they were re-attached to the same individual. Moths were left to recover from the surgery and tested on the following day.

To re-attach the flagella, moths were immobilized by cooling at 3° C for 8 min, followed by 2 min at −20° C. Flagella were quickly attached to the flagellar stump with a small amount of superglue (Loctite Super Glue Gel, Henkel, *Figure 1—figure supplement 1C*). After ensuring that the flagella were properly attached, moths were placed inside a plastic box (10 cm x 10 cm x 8 cm) on a wet tissue paper for 10 min to keep them quiescent and ensure proper reattachment. In case an animal broke the re-attached flagella, a spare one of similar size was used to repeat the re-attachment procedure. Moths were then allowed to recover for a day, before being used in experiments.

We noticed that the flagella lost moisture once re-attached. To quantify the reduction in weight due to moisture loss, we weighed a set of flagella directly after surgery and a few days later when they had dried. Dry flagella had statistically significantly lower weights than freshly ablated flagella (moist: 1.2 ± 0.2 mg, dry: 0.4 ± 0.3 mg; median and inter-quartile range, Wilcoxon rank sum test, z-value = −5.915, p<0.001). We could not determine the weight of the glue used for reattachment, but it is unlikely to exceed the difference between dried and moist flagella, considering the tiny amount of glue used.

To obtain a general idea of the weight ratios of the antenna, head and body of individual hawk-moths, we measured these quantities in six freshly sacrificed animals (three male, three female). The resulting average head:antenna ratio was 3.38:1 ± 0.68 standard deviation, with a weight of 10.5 ± 1.4 mg for the head and 3.1 ± 0.31 mg for the two antennae. The animals weighed an average of 252 ± 66 mg.

## Experimental setup

We used a robotic flower assay as our experimental setup. This assay was first pioneered by *Farina et al. (1994)* and *Sponberg et al. (2015)*, also used in *Stöckl et al. (2017a)*. A flight cage (of the same size as the holding cage) was lined with soft muslin cloth and covered with black cloth on the outside, on three sides, while the front and top were sealed with Perspex windows to allow filming. An artificial flower (48 mm in diameter, on a 140 mm stalk) at the centre of the flight cage, with a nectary (opening of 8.3 mm diameter) filled with 10% sucrose solution, could be moved side-ways (in arcs around the central pole). The position of the flower was controlled by a stepper motor (0.9 degree/step resolution, 1/16 microstepping, Phidgets, Inc.). The motor was interfaced using the Phidget21 MATLAB library (https://www.phidgets.com/docs21/Language_-_MATLAB) with custom written code shared by Simon Sponberg (*Sponberg et al., 2015*). In short, we set the position of the motor using the 'CPhidgetStepper_setTargetPosition' command of the 'phidget21' library according to the trajectory of the stimuli (*Source Data 1*) The cage was illuminated from above with an adjustable white LED panel and diffuser (CN-126 LED video light, Neewer, dimensions: 7.9 × 15.8 cm, 126 individual LEDs, colour temperature: 5400K). The light intensity was set to 3000 lux for the bright light condition and 30 lux for the dim light condition (measured with a Hagner ScreenMaster, B. Hagner AB, Solna, Sweden, at the position of the artificial flower). In addition, two 850 nm IR LED lights (LEDLB-16-IR-F, Larson Electronics) provided illumination for the infrared-sensitive high-speed video cameras (MotionBLITZ EoSens mini, Mikrotron) used to film the flower and moths. Videos were recorded at 100 fps, allowing us to record sequences of up to 28 s, which were required for our analysis of flower tracking. One camera was placed on top of the cage to film the flower and moth from above during all tests. For experiments with the stationary flower, a second camera

providing a rear view was placed on a tripod outside the experimental cage, at approximately 30 cm distance from the artificial flower.

## Behavioral experiments

Eclosed moths were taken from their holding cage and placed in small individually marked cardboard boxes, in which they would be held between trials. For the duration of the experiment, moths were only given access to sucrose from the artificial flower during trials in the experimental cage. A single hawkmoth at a time was introduced into the experimental cage.

We performed two sets of experiments: in the first one, we filmed the moth's approach to and hovering at a stationary flower with both the top and the rear camera. In the second one, we filmed the moth tracking a moving flower using only the top camera. In this set of experiments, we started moving the artificial flower once the moth began to feed from it. We used two different types of movements, the 'sum-of-sines' stimulus and the 'chirp' stimulus, in the same flight bout. The first 16 s of the sequence thus comprised of the pseudo-random sum-of-sine stimulus composed of the following 14 frequencies, which were prime multiples of each other to avoid harmonic overlap: 0.5, 0.7, 1.1, 1.3, 1.7, 1.9, 2.3, 2.9, 3.7, 4.3, 5.3, 6.1, 7.9, 8.9 Hz. High frequencies had lower amplitudes and vice-versa, to assure equal velocities at all frequencies (*Figure 1D* and *Source Data 1*). The sum-of-sines stimulus was followed by a brief stationary phase of 0.5 s, and then the 11 s lasting chirp stimulus with fixed movement amplitude of 10.4 mm and frequencies increasing over time from 0 up to 7.3 Hz (*Figure 1D* and *Source Data 1*). To avoid startling the animals, we did not initiate the movement of the flower abruptly at full amplitude, but rather slowly ramped up (and ramped down) the amplitude of the stimuli over half a second before and after the 'sum-of-sines' and the 'chirp' stimulus. We excluded these portions of the stimulus from our analysis: we extracted 10 s of the flight path with the 'sum-of-sines' stimulus for analysis, 9.5 s from the 'chirp' stimulus, always starting at the same position of the stimulus for all animals.

The protocols were similar for both sets of experiments. Each individual was tested six times: in three antennal conditions (intact *control*, *ablated* and with *reattached* flagella), and in two light intensities (3000 and 30 lux). Because *M. stellatarum* were less motivated to fly in dim light, we first tested the moths in dim light, when they were hungriest and had the highest motivation to forage, and in bright light (3000 lux) later the same day. If a moth did not track both the sum-of-sines and the chirp stimulus (or the stationary flower for at least 6 s), we repeated the test the next day, until a full set of data was collected and the experiment moved on to the next condition. This experimental strategy gave flagella ablated (and re-attached) moths a chance to adapt to their altered mechanosensory feedback, and practice flying and tracking the flower on several days before succeeding. Indeed, our observations suggest that hawkmoths learned to adjust their flight to the lack or change of mechanosensory feedback, as the initial flight attempts of many flagella ablated (and to a lesser degree re-attached) moths showed more severe impairments than consecutive attempts.

## Datasets

Our final datasets include only individuals that tracked the flower in all three antennal conditions in both light intensities. We used six individuals for the experiment with the stationary flower, performed one trial of 6 s at the stationary flower in each condition, and 12 different individuals with the moving flower, which performed one trial (comprised of the sum-of-sines and chirp stimulus) in each condition. Thus, the data analysis of the stationary and moving flower experiments (*Figures 2–4*) has a balanced design, with paired measures for all three antennal conditions and the two light intensities. Moreover, we characterized the general behaviour of all hawkmoths that were part of the moving flower experiment, including those that did not complete all antennal conditions and light intensities, and thus were not included in any further analysis. Thus, the general behaviour scores (*Table 1*) does not comprise a balanced design, and contains repeated measures (which were accounted for in the statistical analysis, see *Data Analysis* below).

## Data analysis

The positions of the flower and the hawkmoth were digitised from the videos using the DLTdv5 software for MATLAB (*Hedrick, 2008*; *Dyhr et al., 2013*). In experiments with stationary flowers, both the approach and the stationary hovering were digitised, whereas in experiments with moving

flower, only sequences during which the proboscis of a moth was in contact with the nectary were rated as 'tracking' and digitized (as in *Sponberg et al., 2015*; *Stöckl et al., 2017a*). In the top view, a point on the flower, and a reliably identifiable point on the pronotum of the moth were used for reference. From the rear view videos, we used the centre of the nectary, the centre of the pronotum and the centre tip of the abdomen (*Figure 3*).

## General behaviour

We characterised the general behaviour of all hawkmoths in the moving flower experiments, including those that did not complete all antennal conditions and light intensities, and thus were not included in any further analysis. We classified their behaviour into three different categories (*Table 1*): non-flying (animals which would not take off after 5 min in the experimental cage), flying (animals which flew but would not feed from the flower) and tracking (animals feeding from and tracking the flower, at least partially). We used multinomial logistic regression (package mlogit v0.2–4: (*Croissant, 2013*) to model the rates of one of the three behaviours (*non-flying*, *flying*, *tracking*, *Table 1*) as a function of antennal condition and light intensity, including the identity of individual moths as a random factor (*Supplementary file 1*).

## Stationary flower experiments

To compare the stability of hovering flight between the different antennal conditions and light intensities, we analysed the position of the thorax and abdomen for a 6 s interval of hovering at the flower nectary during feeding (given perfect hovering, the thorax should retain a stable position, because the flower was immobile). We quantified the amplitude of thorax and abdomen movements across different movement frequencies by Fourier transforming their position over time (*Figure 2B, C*). To assess the effect of antennal condition across frequencies, we applied a linear mixed-effects model (*Bates et al., 2015*) with antennal condition, frequency and their interaction as fixed effects and individual identity as a random effect on the log-transformed magnitudes of body movement. We confirmed that the full model did explain the variance better than reduced versions of the model (likelihood ratio test) before performing post-hoc comparisons using the 'lmerTest' package in R (*Kuznetsova et al., 2017*).

To compare these measures across light intensities, we calculated the difference between the log-transformed magnitude spectra of thorax and abdomen position in bright and dim light for each antennal condition (*Figure 4C,D*). We then compared these using general linear models of the same form as above.

## Sum-of-sines movement

We used system identification analysis (*Cowan et al., 2014*) to characterise hawkmoth flower tracking performance. This analysis is possible because the sum-of-sines stimulus fulfils the requirement of linearity, that is it generates the same flower tracking performance at different amplitudes and phase relationships (see Supplement, *Stöckl et al., 2017a*). Hawkmoth flower tracking can be described by two components: gain and phase (*Farina et al., 1994*; *Sponberg et al., 2015*; *Stöckl et al., 2017a*). Gain relates the amplitude of flower movement to hawkmoth movement (one for perfect tracking), while the phase describes the lead or lag of the hawkmoth with respect to the flower movement (0 for perfect tracking). We used a metric called tracking error ε (*Roth et al., 2014*; *Sponberg et al., 2015*), which incorporates effects of both gain and phase to quantify tracking performance of hawkmoths (*Figure 3B*). It is calculated as the complex distance between the moth's response H(s) and the ideal tracking conditions (gain = 1, phase lag = 0), where s is the Laplace frequency variable:

$$\epsilon(s) = \|H(s) - (1 + 0i)\| \tag{1}$$

A tracking error of 0 means perfect tracking (comprising a gain of 1 and a phase lag of 0), while the tracking error is one if the hawkmoth and flower movement are uncorrelated (e.g. when either the hawkmoth remains stationary and the flower moves, or vice versa). We calculated average tracking errors and their confidence intervals within antennal conditions by averaging data in the complex plane, to avoid artefacts resulting from separating gain and phase components when transforming them and averaging in the non-complex plane (see *Stöckl et al., 2017a* for discussion).

Since our tracking error metric is a complex value and was only transformed into the non-complex plane after averaging across individuals, it is not straightforward to find appropriate statistical tests to compare tracking error (as well as gain and phase) across antennal conditions and light intensities. Linear mixed effects models would be well suited, but complex data might not fulfil all of the assumptions these models are based on. Lacking an alternative, we had to rely on these tests, as did previous studies with the same approach (*Roth et al., 2016*; *Sponberg et al., 2015*) to compare the effect of antennal conditions across frequencies as fixed effects, including individual identity as a random effect. We are confident that overall trends identified as statistically significant by these models are indicative of biologically relevant effects, but advice caution when interpreting differences in significance at individual flower movement frequencies isolated from the overall trend.

To compare tracking performance across light intensities, we calculated the difference in the flower frequency at which tracking error reached one for both dim and bright light intensity within antennal conditions (similar to *Stöckl et al., 2017a*), and compared these across antennal conditions (*Figure 4A*). For statistical comparisons, we used the Friedman test, which is a non-parametric test that accounts for repeated measures.

## Chirp movement

The chirp stimulus does not fulfil the linearity criterion, because it does not generate the same flower tracking performance at different amplitudes and phase relationships, but rather contains a saturation non-linearity which makes it increasingly harder for moths to track the flower with increasing flower frequency. Thus, the system identification analysis we used for the sum-of-sines stimulus could not be applied (*Roth et al., 2014*).

We therefore determined the flower frequency, at which each individual lost proboscis contact with the flower (i.e. failed at tracking the flower) as a measure of flower tracking performance across frequencies. This measure gave an absolute cut-off frequency at which moths could no longer track the oscillating flower. Because this data was non-parametric and included repeated measures, we used a Friedman test to compare the paired data. To compare the tracking performance across light intensities, we calculated the difference in flower frequency between dim and bright light at which each individual in each antennal condition stopped tracking the flower. These differences between light conditions were then compared across antennal conditions using a Friedman test to retain information about the paired data (*Figure 4B*).

In order to resolve the finer differences in flower tracking performance at the chirp stimulus between the three antennal conditions, we also performed a frequency analysis, and calculated the amplitude spectrum of the responses, as well as the cross power spectrum density and phase relationship between moth and flower tracks (*Figure 3—figure supplement 3*).

## Acknowledgements

We thank Merry and Leigh Foster for help with capturing the parental moths in France, Michael Pfaff and Joaquin Goyret for performing pilot experiments, Simon Sponberg for installing our robotic flower set-up and for critical comments on the manuscript, Marie Dacke for allowing us to use two high speed cameras, David O'Carroll for inspiring discussions, Karin Nordström for valuable comments on the manuscript, and Eric Warrant for financial support of AS.

## Additional information

### Funding

| Funder | Grant reference number | Author |
| --- | --- | --- |
| Vetenskapsrådet | VR621-2012-2212 | Almut Kelber |
| Knut och Alice Wallenbergs Stiftelse | | Almut Kelber |
| Carl Tryggers Stiftelse för Vetenskaplig Forskning | 15:108 | James J Foster |

| Erasmus+ | Erasmus Mundus Scholarship | Ajinkya Dahake |
| Air Force Office of Scientific Research | FA2386-11-1- 4057 | Sanjay P Sane |

The funders had no role in study design, data collection and interpretation, or the decision to submit the work for publication.

### Author contributions
Ajinkya Dahake, Data curation, Formal analysis, Investigation, Writing—review and editing; Anna L Stöckl, Data curation, Formal analysis, Visualization, Methodology, Writing—original draft, Writing—review and editing; James J Foster, Formal analysis, Writing—review and editing; Sanjay P Sane, Conceptualization, Supervision, Funding acquisition, Writing—review and editing; Almut Kelber, Conceptualization, Resources, Formal analysis, Supervision, Funding acquisition, Methodology, Project administration, Writing—review and editing

### Author ORCIDs
Anna L Stöckl (iD) http://orcid.org/0000-0002-0833-9995
James J Foster (iD) http://orcid.org/0000-0002-4444-2375
Sanjay P Sane (iD) http://orcid.org/0000-0002-8274-1181
Almut Kelber (iD) http://orcid.org/0000-0003-3937-2808

### Decision letter and Author response
Decision letter https://doi.org/10.7554/eLife.37606.033
Author response https://doi.org/10.7554/eLife.37606.034

## Additional files

### Supplementary files
• Source data 1. Contains the 'sum-of-sines' and 'chirp' stimulus used in this study as MATLAB arrays, as well as MATLAB scripts to generate the stimuli.
DOI: https://doi.org/10.7554/eLife.37606.017

• Supplementary file 1. Results of the statistical models assessing the effect of antennal treatment and light intensity on the proportion of different behaviours in the flight cages. The behaviour of each animal was classified into the following categories: *no flight*, *flight* (but no tracking of the flower), and *tracking*. Some moths were tested multiple times to collect the necessary tracking data, and thus have contributed multiple trials to this dataset. Statistical comparisons were performed using multinomial regression including the identity of individual moths as a random factor, to model the rates of one of the three behaviours as a function of antennal condition and lighting. As no significant interaction between antennal condition and light intensity was found, the fixed effects of the fitted model took the form: behavioural category (no flight, flight, tracking)~antennal condition+light intensity. All statistical results are expressed in relation to the probability of observing the *no flight* behaviour in the control condition in bright light.
DOI: https://doi.org/10.7554/eLife.37606.018

• Supplementary file 2. Results of the statistical models assessing the effect of antennal treatment on thorax jitter in the stationary experiment in bright light (*Figure 2B*). A general linear model was constructed with antennal treatment and frequency (binned to the logarithmic scale) as factors: log (response)~antennal condition * frequency +1|individual.
DOI: https://doi.org/10.7554/eLife.37606.019

• Supplementary file 3. Results of the statistical models assessing the effect of antennal treatment on abdomen jitter in the stationary experiment in bright light (*Figure 2C*). A general linear model was constructed with antennal treatment and frequency (binned to the logarithmic scale) as factors: log (response)~antennal condition * frequency +1|individual.
DOI: https://doi.org/10.7554/eLife.37606.020

• Supplementary file 4. Results of the statistical model assessing the effect of antennal treatment on flower tracking performance with the sum-of-sines stimulus in bright light (*Figure 3B*): a general linear model was constructed with antennal treatment and frequency (binned to the logarithmic scale) as factors: log(response)~antennal condition * frequency +1|individual.
DOI: https://doi.org/10.7554/eLife.37606.023

• Supplementary file 5. Results of the statistical model assessing the effect of antennal treatment on flower tracking performance with the chirp stimulus in bright light (*Figure 3D*): a Friedman test was performed, with a Tukey-Kramer post-hoc comparison correction for multiple comparisons.
DOI: https://doi.org/10.7554/eLife.37606.024

• Supplementary file 6. Results of the statistical model assessing the effect of antennal condition on the difference in flower tracking error between light conditions with the chirp stimulus (*Figure 4B*). A Friedman test was performed, with a Tukey-Kramer post-hoc comparison correction for multiple comparisons.
DOI: https://doi.org/10.7554/eLife.37606.028

• Supplementary file 7. Results of the statistical model assessing the effect of antennal condition on the difference in thorax stability during hovering between light conditions (*Figure 4C*). A Friedman test was performed, with a Tukey-Kramer post-hoc comparison correction for multiple comparisons.
DOI: https://doi.org/10.7554/eLife.37606.029

• Supplementary file 8. Results of the statistical model assessing the effect of antennal condition on the difference in flower tracking error between light conditions with the sum-of-sines stimulus (*Figure 4A*). A Friedman test was performed, with a Tukey-Kramer post-hoc comparison correction for multiple comparisons.
DOI: https://doi.org/10.7554/eLife.37606.027

• Supplementary file 9. Results of the statistical models assessing the effect of antennal treatment on thorax jitter in the stationary experiment in dim light (*Figure 2—figure supplement 2A*). A general linear model was constructed with antennal treatment and frequency (binned to the logarithmic scale) as factors: log(response)~antennal condition * frequency +1|individual.
DOI: https://doi.org/10.7554/eLife.37606.021

• Supplementary file 10. Results of the statistical models assessing the effect of antennal treatment on abdomen jitter in the stationary experiment in dim light (*Figure 2—figure supplement 2B*). A general linear model was constructed with antennal treatment and frequency (binned to the logarithmic scale) as factors: log(response)~antennal condition * frequency +1|individual.
DOI: https://doi.org/10.7554/eLife.37606.022

• Supplementary file 11. Results of the statistical model assessing the effect of antennal treatment on flower tracking performance with the sum-of-sines stimulus in dim light (*Figure 3—figure supplement 1*): a general linear model was constructed with antennal treatment and frequency (binned to the logarithmic scale) as factors: log(response)~antennal condition * frequency +1|individual.
DOI: https://doi.org/10.7554/eLife.37606.025

• Supplementary file 12. Results of the statistical model assessing the effect of antennal treatment on flower tracking performance with the chirp stimulus in dim light (*Figure 3—figure supplement 4*): a Friedman test was performed, with a Tukey-Kramer post-hoc comparison correction for multiple comparisons.
DOI: https://doi.org/10.7554/eLife.37606.026

• Supplementary file 13. Results of the statistical model assessing the effect of antennal condition on the difference in abdomen stability during hovering between light conditions (*Figure 4D*). A Friedman test was performed, with a Tukey-Kramer post-hoc comparison correction for multiple comparisons.
DOI: https://doi.org/10.7554/eLife.37606.030

• Transparent reporting form
DOI: https://doi.org/10.7554/eLife.37606.031

### Data availability

All data generated or analysed during this study are included in the manuscript and supporting files. Source data files have been provided for Figures 2 and 3, as well as Figure 2-figure supplement 1, Figure 2-figure supplement 2 and Figure 3-figure supplement 1.

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
