## [Decision Letter]

Thank you for submitting your article "The roles of vision and antennal mechanoreception in hawkmoth flight control" for consideration by *eLife*. Your article has been reviewed by three peer reviewers, including Ronald L Calabrese as the Reviewing Editor and Reviewer #1, and the evaluation has been overseen by Eve Marder as the Senior Editor. The following individual involved in the review of your submission has agreed to reveal their identity: Noah Cowan (Reviewer #2).

The reviewers have discussed the reviews with one another and the Reviewing Editor has drafted this decision to help you prepare a revised submission.

Summary:

In this manuscript, the authors present an analysis of the interaction of antennal mechanosensory and visual input in the stabilization of hovering flight in a diurnal moth. The analysis compares three antennal conditions (intact control, antennae removed, antennae reattached) and two light conditions mid-range and low, each with two stimuli stationary flower feeding, and oscillating flower feeding consisting of two movements: chirp and sum of sines. They find that for both stimuli in the mid-range lighting, moths show degraded tracking at higher frequencies with removal of antennae and recover somewhat when antennae are reattached, compared to control (intact antennae). In low lighting, all three conditions behave similarly poorly, even at low frequencies. These results are consistent with the emerging conclusion that visual input is essential for moths to identify and track the flower movement relative to their own position – antennal mechanoreceptors cannot provide that required information but are necessary to support fast flight maneuvers. Vision and mechanoreception thus act in different frequency domains and do not compensate. These studies complement those showing similar interactions between haltere mechanoreception and vision in dipterans (most notably *Drosophila*) by extending them to the greater number of insect orders that lack halteres and thus make the findings of wide interest.

Essential revisions:

The expert reviews are provided, which will require rewriting and some new analyses. Important points include:

1) All three reviewers agreed in consultation that a control diagram is needed to help the reviewers contextualize the findings and point the way forward for further mechanistic analyses.

2) In discussion among the reviewers, there was some concern about the phase analysis and the reviewers discussed whether cross correlation would be a better strategy. However they concluded that cross correlation could be tricky with a chirp or sum of sines. Indeed, cross-correlation should be done at a given frequency. If these data are available then a cross correlation might be in order.

3) The expert reviews provided should all be addressed; they are detailed but consistent and complementary.

Reviewer #1

Concerns

1) I found it confusing that the essential idea or hypothesis that visual input is essential for moths to identify and track movement relative to their own position regardless of frequency is not made clear up front. The presentation could be made a lot clearer, if the contrast between mid-range and low light was presented first for the intact condition. Perhaps this problem will be ameliorated by the inclusion of a control diagram.

Reviewer #2:

Concerns

The most significant issue is the lack of a clear interpretation of the results. The results of this paper are quite interesting; the removal of a "postural" self-motion (similar to proprioception or vestibular) feedback system, the antennae, affects tracking of the exogenous motion of a flower. Why is this so interesting? Because of the subtle but evidently important interaction between these distinct feedback loops (diagram in Author response image 1):

One could explain (at least) qualitatively this interaction, as depicted in a graphical feedback control model such as the one above. It seems interesting that ablating the antenna disrupts the inner loop of the control system, making the outer loop (vision and mechanoreception from proboscis) not as effective at tracking. How does the modulation of these inner-loop dynamics (based on ablation) hinder outer-loop control? This is ripe for interesting computational modeling – such modeling itself could be saved for future work, but the description of this problem, which now is only vaguely hinted at, would elevate the paper substantially. The diagram above is just a rough cut and needs to be fleshed out, but I believe it to be a reasonable stab at the topology of the feedback system in question and if the authors agree, I encourage them to adapt and include something similar. Some possible issues with my above diagram that will require greater thought by the authors:

– the summing junction after the antennal and visuo-mechanosensory blocks is a simplifying assumption,

– before the plant perhaps there should be some sort of CNS integration

– Probably it is a +/- but the second junction may be better as rectangle labeled

"multisensory integration" where the inner-workings are left as future work.

In addition to that we have a number of detailed comments.

Detailed comments:

1) The authors use the word "significantly" even for data that, while statistically significantly different, are not that different from a controls engineering point of view. This is most notable in the discussion of Figure 3B, where the tracking errors of the re-attached and ablated moths were "significantly different" but compared to controls, they were quite similar.

2) Sometimes ablated/reattached moths perform more like controls, and sometimes they perform more like antennectomized moths. Can this be fleshed out a bit? For example as in point #1, the reattached and antennectomized moths performed similarly, but in the chirp task, control and re-attached were more similar.

3) It is unclear that how many trials are performed for each individual animal in each different case. It seems "one set of data", but it should be described in the paper. If you did not do more than one trial per animal (e.g. multiple passes of sums-of-sines), why not? If you did, how many did you do, how did you perform averaging, etc.? As long as that can be clarified, the results are compelling and seem to support the overall claims of the paper.

4) The paper has no citations to the Cohen lab papers that include models of haltere-based flight control in flies over the last 8 years that include roll, pitch, and yaw perturbations. There are crucial experimental differences; you perturb target motion (tracking task) and they provide a mechanical perturbation (i.e. a disturbance), so the experimental topology is different but perhaps their data could give you some insights into the inner-loop control structure?

5) The paper assumes that is the proboscis mechanoreception has "little or no" feedback contribution. Recent and crucial work by Roth et al. that the authors site quite clearly indicates otherwise for a related species of moth. This manuscript does use data in dim and bright light in the phase before proboscis contact with the nectary for a stationary flower (Figure 2—figure supplement 1), in which case this assumption seems valid, but mechanoreception is known to play a large role in tracking a moving flower (Roth et al.). That said, I'm not sure why this rather dubious assumption is needed.

6) Tracking error decreases significantly after reaching to its peak (Figure 3B, Figure 3—figure supplement 1). It is not intuitively clear why or how tracking error would decrease at higher frequencies. Is there any explanation for that? I have some concern that phase lags may have "wrapped" but that the analysis performed didn't identify that wrapping. Could that be possible? See especially Figure 3—figure supplement 1, last row.

7) As shown in Figure 1D, the power for chirp stimulus is much larger than the power for sum-of- sines. In subsection “Chirp movement”, it is mentioned that "chirp stimulus does not fulfill the linearity criterion".

Was this measured quantitatively? If so, how? Maybe if chirp had similar amplitude as the sum-of-sines then it would be linear. Is there a reason for chirp stimulus to have such large amplitude (and consequently high power)?

8) The frequency response could be calculated using the data from the chirp stimulus (by using cross spectral density and power spectral density). Also, in Figure 3C, using a second x-axis to show the frequency would be helpful.

9) The result for chirp stimulus tracking for control and reattached are very similar (Figure 3C, 3D). What is the explanation for this high similarity?

10) In the Results section paragraph four, the sentence is vague: "abdominal jitter of moths with re-attached flagella differed significantly from control moths at only one frequency (1.66 Hz)". What's the explanation for that? Why only one frequency? Is this meant to be the same frequency that is listed as 1.7 Hz in subsection “Behavioural experiments”?

11) In paragraph two of subsection “Flagella ablation reduces flower tracking performance at high frequencies in hawkmoths”, based on Figure 3B, the tracking error doesn't look significantly higher. Does it mean statistically significant?

12) Figure 4C, 4D: The plots in three colors have overlapped each other so much that they are unclear. Other methods can be used to show the change of noise in different frequencies.

13) In Figure 4C, it is unclear why the dimension of amplitude is shown as Hz.

Reviewer #3:

Concerns

– A parallel is made by authors between halteres role in Dipteran and flagella in hawkmoth. However, it is surprising that the work of Itai Cohen group was not cited in the Introduction (paragraph two). Cohen was probably the first to measure disturbance rejection in free flying fruitfly around the three axes of rotation (roll, pitch and yaw).

It is worth noting that halteres have an essential role in the stabilization of the fly, which is not the case in hawkmoth without flagella: the tracking accuracy is degraded but the flight is still stable in hawkmoth. Only the jitter seems to be higher without flagella but the flight remains stable in hover: the tracking seems to be slower indeed. I do not see any instability according to the definition proposed by control engineers. For example in figure 2, hovering flight with ablated flagella is not instable but less accurate. I suggest authors mention more a more accurate control than an instable control.

– Introduction paragraph four: it is mentioned that antennal mechanosensors play a key role but what kind of role? Authors should add a block diagram to clarify their model and to show clearly the closed-loop control the hawkmoth position:

– the tracking error between the insect and target position will be shown

– the inner loop based on the antenna block could be shown with respect to the control of the head in body orientation.

As the antennal ablation does not introduce instability but less accurate visual tracking, authors could discuss the fact that antennal mechanoreceptors could act as the prosternal organs in fly to allow the animal to measure the orientation of its head with respect to the insect body (head in boy orientation). As suggested by Viollet and Zeil, 2013, prosternal organs may be involved in a mechanoreceptive feedback on head position relative to the thorax. Antennal mechanoreceptors in hawkmoth could play a similar role: they could just improve the control (accuracy) of the head orientation (gaze).

– Subsection “Sum-of-sines movement”: about the tracking error, please clarify why you used this particular metric defined by equation 1.

– What kind of algorithms and software were used to estimate the gain and phase? I recommend having an open access to the programs (code).

[Editors' note: further revisions were requested prior to acceptance, as described below.]

Thank you for resubmitting your work entitled "The roles of vision and antennal mechanoreception in hawkmoth flight control" for further consideration at *eLife*. Your revised article has been favorably evaluated by Eve Marder (Senior Editor), a Reviewing Editor, and two reviewers.

The manuscript has been improved but there are some remaining issues that need to be addressed before acceptance, as outlined below:

The authors have done an excellent job of addressing the reviewer concerns except in two points.

Essential Revisions:

– Adding the detailed block diagram figure:

The expert reviewers are in strong agreement in consultation that the present control diagram is inaccurate. They recommend that the attached block diagram topology with inner loop be adopted. See the detailed reviews for their reasoning. If the authors insist that a 'simpler' diagram is needed, then the attached block diagram with NO inner loop may be used. Again see the detailed reviews for their reasoning.

– Phase analysis:

There are serious concerns about the phase analysis that were not adequately addressed in revision. See the detailed reviews for how these concerns can be addressed.

Reviewer #2:

Block diagram.

The authors introduce a different block diagram than what was suggested originally, they mentioned that they don't want to limit their conclusion to flower tracking so they introduce a more general block diagram. They want to mention the whole flight control in their block diagram, "not just tracking of a moving flower". The authors' new block diagram (Figure 1A) is unfortunately topologically incorrect. A "postural perturbation" is not defined in this paper anywhere that I can find but let us assume that when the diagram is "reduced" to the present paper, it is a moving flower (since that is all the paper addresses in the way of perturbations, although I wouldn't ever refer flower motion as a "postural perturbation" because it is a sensory perturbation that may lead to postural sway but only indirectly through the visuomotor system; not incorrect but not very clear either).

In any case, the moth's motion is subtracted from the flower motion before going into the visual system but self-motion is *not* subtracted from flower motion before going to the antennae (unless you are somehow modeling the antenna as a wind sensor which is unlikely). Since I am revealing myself in this review I would be happy to discuss this point further but as drawn this diagram does not make sense. The self-motion feedback to the antenna is grounded and is not subtracted from the sensory feedback. A perfectly acceptable approach would be to remove the subtraction bubble altogether, and draw it in the way that I have suggested.

(You would need to re-do the graphics inside the Sensory Input block to not give the sense that self-motion feedback only goes to the antennae.) The left arrow could be something like "Exogenous perturbation" and the return arrow could be "Self-motion feedback" or, maybe "Exafferent perturbation" for the left incoming error and "Reafferent feedback" for the return path.

Phase lag

Second, the problem with your currently unwrapped phase is that it is -π/2 at 10Hz, which means that the time lag between the input stimulus and the moth motion is 25ms, which seems extremely fast. I've never seen it in any visuomotor control paper in any species (from external motion to animal motion). If you look for example at Roth et al., 2016 he shows roughly 3π/4 to 2π (almost 360^o^) of phase lag, which corresponds to something like 75ms to 100ms which is a lot more sensible. This is very much in line with the feedback delay estimated by Sponberg, 2016 in their "Luminance dependence" paper. It is not possible it is only 25ms of total time lag (delay + low-pass mechanical phase lag). I don't see a problem with the approach but it is a completely unbelievable result. There are many possible sources of this.

One problem I've had is when I have data streams from different sources that get temporally offset or that temporally drift. This can introduce leads / lags. Another more likely possibility is that the roll off is so fast that your attempt to unwrap just misses it. See the incredible roll off in Eatai Roth's recent PNAS paper. The fastest latency I've seen estimated is Dyhr et al., 2013 (hawkmoth abdomen) which was 41ms but keep in mind that was just to the abdomen not the entire flight behavior, and it is quite possible that the high-pass "lead" filter helped mask some of that delay. Even still at 10Hz, the phase lag was π (180^o^) from stimulus to abdominal movement. The flight mechanics would surely introduce more phase lag.

Based on the error analysis in the complex plane (which I never really doubted even if I got a bit confused at one point – The explanation re: reduced tracking error at high frequencies is reasonable and I should have realized it before.) I don't think this will affect the main conclusions of the paper but it really does need to be addressed.

Reviewer #3:

There are just three points I would like to address again:

– the new block diagram is not accurate enough. I suggest separating the vision block from the antennal mechanosensors. A visual error can result from a difference between the moth's head orientation and the flower's position. These three signals (head's orientation, flower position and visual error) must be indicated on the diagram. This visual error can then be sent into the vision block, the output of which can be sent to the central integration block. As the mechanosensors seem to play a major role in the stabilisation of the moth, I suggest inserting this block in an inner loop with the motor system block. I agree that this point needs further experiments to determine precisely the function of the mechanosensors.

– Would it be possible that antennae act as a lead compensator (derivator) as the oscillations (jitter) are reduced? This point could be addressed in the discussion.

– Discussion about the role of the antenna to stabilise the head on the basis of the work of Viollet and Zeil (JEB paper) is not included.

[Editors' note: further revisions were requested prior to acceptance, as described below.]

Thank you for resubmitting your work entitled "The roles of vision and antennal mechanoreception in hawkmoth flight control" for further consideration at *eLife*. Your revised article has been favorably evaluated by Eve Marder (Senior Editor), a Reviewing Editor, and one reviewer.

The manuscript has been improved but there are some remaining issues that need to be addressed before acceptance, as outlined below:

This is an unusual case where the reviewer's rationale for a rather minor required revision requires a rather long argument. Basically this reviewer calls for a caveat to be added to the Discussion in a short paragraph (or a few sentences to an existing paragraph). This caveat will not change the impact of the work, but will help the reader understand the rather remarkable tracking ability of this moth. Revision can be very swift and will not require re-review.

Reviewer #2:

The authors have done a remarkable job addressing my comments. I particularly appreciate their effort on technical issues such as providing extra data, revised PDFs, etc. Scientifically, I am convinced by the arguments in the revised manuscript; the updated block diagram and the other more minor issues I raised have also been addressed.

The one remaining issue about which I found myself concerned was the issue of phase lag. 90^o^ at 8.9Hz is really quite extraordinary in the animal kingdom and having done system ID on moths, fish, humans, cockroaches, fruit flies, and even non-moving system ID on the jamming avoidance response in electric fish (where there is no "inertial low pass filter"), I've never seen such a short delay on a sensorimotor feedback loop except maybe from some work on haltere feedback. But not only have I been able to recapitulate the results based on their uploaded MATLAB data files, but also was able to reproduce the results from the raw image data, performing my own image tracking. In fact, in that video the phase lag at 8.9Hz is a mere 66^o^(with excellent SNR), corresponding to a mere 20ms visuo-movement response (total phase lag, including delay and mechanical phase lag). I just simply don't believe that is possible.

The authors claim that the animal is smaller and has a higher wingbeat frequency and therefore 'in-cycle' control would mean low phase lag, but I do not believe the synapses are any faster in this moth than in any other insect – and not even a fruit fly with hundreds of wingbeats per second can respond that fast to visual perturbations (fastest shown about 30ms I believe).

However, I do think I have a possible explanation, which is that there is a direct, mechanical coupling between the flower and the head of the moth via the proboscis. It is the only thing that I can see from the videos that could explain this extraordinary response. (Note that I measured to the tip of the head, not the thorax, when measuring the 20ms lag at 8.9Hz). That said, I think it is clear from the video that there is still a strong sensorimotor component to the behavior and the comparison being made in the paper includes this mechanical coupling for both antenna intact and antenna-ablated conditions. So I do not think that the possible mechanical coupling undermines the results in anyway, since the coupling was present in both conditions.

However, I would appreciate if the authors could acknowledge that these phase lags are unusually short (no known examples in the literature), and that some mechanical coupling may be playing a role. This can be a discussion point and needn't be a major point. It should also say that any such mechanical coupling would be present in both groups (intact vs ablated antenna), and doesn't impact the main findings of the paper.

As an extra step I looked at the bode plots from flower to thorax and flower to abdomen. The thought is that "yanking" the proboscis to the left and right might rotate the body quickly (and therefore show a low phase lag to the rostral end of the animal) but may not move the thorax directly. The moth seems to be rotating around the thorax. I did see much greater phase lags – on the order of 50ms – when I looked at the flower-to-thorax (near the rear of the thorax) transfer function.

---

## [Author Response]

Essential revisions:The expert reviews are provided, which will require rewriting and some new analyses. Important points include:1) All three reviewers agreed in consultation that a control diagram is needed to help the reviewers contextualize the findings and point the way forward for further mechanistic analyses.

We agree with the reviewers that a control diagram is beneficial to visualize the findings, and to support future control theoretical analyses, and have therefore added it to Figure 1.

2) In discussion among the reviewers, there was some concern about the phase analysis and the reviewers discussed whether cross correlation would be a better strategy. However they concluded that cross correlation could be tricky with a chirp or sum of sines. Indeed, cross-correlation should be done at a given frequency. If these data are available then a cross correlation might be in order.

We are a little bit puzzled at the concern with our phase analysis we applied to the sum-of-sines data, as this has been used in similar contexts in a range of previous studies with very similar (if not identical) settings (Sponberg, et al., 2015, Roth, et al., 2016, Stöckl et al., 2017), and has a strong theoretical foundation (Roth, Sponberg, and Cowan, 2014). We have demonstrated how our data fulfils the criteria required for this type of analysis. We have responded to the specific concerns related to the analysis in detailed comments below. Should these not alleviate the concerns, then we would be very grateful for a discussion with the reviewers on how to best improve the analysis of the data.

3) The expert reviews provided should all be addressed; they are detailed but consistent and complementary. All minor comments should be addressed.

Below, we respond to all suggestions point by point, and highlight how we have implemented them in the revised manuscript.

Reviewer #1Concerns1) I found it confusing that the essential idea or hypothesis that visual input is essential for moths to identify and track movement relative to their own position regardless of frequency is not made clear up front. The presentation could be made a lot clearer, if the contrast between mid-range and low light was presented first for the intact condition. Perhaps this problem will be ameliorated by the inclusion of a control diagram.

As the reviewer suggests, we hope that the inclusion of a control diagram in Figure 1 helps to dispel any remaining confusions on this point, and also supports our discussion of the importance of vision for flight control in the introduction.

Reviewer #2:Concerns

*The most significant issue is the lack of a clear interpretation of the results. The results of this paper are quite interesting, the removal of a "postural" self-motion (similar to proprioception or vestibular) feedback system, the antennae, affects tracking of the exogenous motion of a flower. Why is this so interesting? Because of the subtle but evidently important interaction between these distinct feedback loops.* (Diagram in Author response image 1):

One could explain (at least) qualitatively this interaction, as depicted in a graphical feedback control model such as the one above. It seems interesting that ablating the antenna disrupts the inner loop of the control system, making the outer loop (vision and mechanoreception from proboscis) not as effective at tracking. How does the modulation of these inner-loop dynamics (based on ablation) hinder outer-loop control? This is ripe for interesting computational modeling – such modeling itself could be saved for future work, but the description of this problem, which now is only vaguely hinted at, would elevate the paper substantially. The diagram above is just a rough cut and needs to be fleshed out, but I believe it to be a reasonable stab at the topology of the feedback system in question and if the authors agree, I encourage them to adapt and include something similar. Some possible issues with my above diagram that will require greater thought by the authors:– the summing junction after the antennal and visuo-mechanosensory blocks is asimplifying assumption,– before the plant perhaps there should be some sort of CNS integration– Probably it is a +/- but the second junction may be better as rectangle labeled"multisensory integration" where the inner-workings are left as future work.

While we agree with the reviewer’s interpretation, we would like to highlight that it is only covering a part of our findings. We have also shown an impairment in flight control during the approach flight to the flower, as well as during hovering in front of a stationary flower, in addition to the impairments observed while the flower is moving and forcing the moths to perform flight manoeuvres to track it. We therefore conclude that our results show that the postural self-motion feedback is required for all aspects of hawkmoth flight, not just tracking of a moving flower. Thus, tracking of a moving flower is just one specific aspect of the effect we are describing, namely that hawkmoths require feedback from antennal mechanosensors for stable flight.

The other main finding of our study is that postural control through vision and antennal mechanosensation seem to operate in different frequency bands (as they do in flies). Thus, we suggest that both vision and antennal mechanosensation contribute to this postural control loop, and that they are parallel systems, which don’t seem to be able to compensate for each other.

Given our interpretation of the results, the fact that ablating the flagella makes tracking a moving flower less accurate is almost a necessary conclusion – because even if the moths can still perfectly determine the movement of the flower using vision and potential mechanosensory feedback from the proboscis, they cannot follow the movement due to their impaired postural control in the air. This leads to a worse flower tracking performance – not because the sensory information about flower movement is impaired, but because moths with ablated flagella are poorer at stabilizing flight than intact moths.

We have highlighted the roles of vision and antennal mechanosensation for postural control in a control model (Figure 1), which will help the interpretation our results. However, it is not clear how flower tracking and its sensory control interacts with the postural control circuit, and our study was not designed to investigate this question (but to lay the groundwork for the role of antennal mechanosensors in postural control in hawkmoths). Hence, we propose a control diagram that is more general, rather than a specific one that integrates flower tracking and postural control. We feel that a more meaningful model can be proposed once we have more information on the mechanisms of integration between the two control circuits, in a manner that is experimentally testable, which is not currently the case. We therefore agree with the reviewer that this would be a very interesting question for future studies.

In addition to that we have a number of detailed comments.Detailed comments:1) The authors use the word "significantly" even for data that, while statistically significantly different, are not that different from a controls engineering point of view. This is most notable in the discussion of Figure 3B, where the tracking errors of the re-attached and ablated moths were "significantly different" but compared to controls, they were quite similar.

We used the word “significant” in the sense of statistically significant, as this, in our eyes, is an objective way to describe the data. We are aware that this might not always relate to significant differences in control engineering points of view, or in biological terms, but we deemed it the least subjective way to determine differences. We therefore have now rephrased all our uses of the word “significant” to clarify its meaning.

2) Sometimes ablated/reattached moths perform more like controls, and sometimes they perform more like antennectomized moths. Can this be fleshed out a bit? For example as in point #1, the reattached and antennectomized moths performed similarly, but in the chirp task, control and re-attached were more similar.

We agree with the reviewer that this point has to be discussed at more length, and therefore expanded our discussion of this point in the manuscript (Discussion – Flagellar re-attachment improves flight performance). In short: we suggest that flagella re-attachment improves, but does not entirely restore flight performance (postural control) to control levels (as has also been observed in a previous study on hawkmoths, Sane et al., 2007). We discuss reasons for this in the manuscript. One likely reason why we observe slightly different trends between conditions in the different experimental paradigms is that they challenge the postural control system of the moths to different degrees. All experimental paradigms though show the same basic result: flagella re-attachment improves flight performance, but does not entirely restore it. This change in flight performance occurs mainly at higher frequencies of movement.

3) It is unclear that how many trials are performed for each individual animal in each different case. It seems "one set of data", but it should be described in the paper. If you did not do more than one trial per animal (e.g. multiple passes of sums-of-sines), why not? If you did, how many did you do, how did you perform averaging, etc.? As long as that can be clarified, the results are compelling and seem to support the overall claims of the paper.

We have now compiled the information about the number of animals and number of trials used in a section of the Materials and methods labelled “Datasets” and hope this makes it clearer. To summarize: the results shown in Figures 2-4 and related supplementary material were obtained from a paired experimental design: 6 animals were used in the stationary flower experiment, which performed one trial at the flower in each antennal condition and each light intensity. 12 different animals were used in the moving flower experiment, which similarly performed one trial (comprising the chirp and sum-of-sines stimulus) in each antennal condition and light intensity. Only the behavioural scores in Table 1 do not represent a balanced design. They were obtained from all animals that participated in the moving flower experiments, including those that only performed in a single or a few conditions – which has been considered in the statistical analysis (for details see Materials and methods, Datasets).

4) The paper has no citations to the Cohen lab papers that include models of haltere-based flight control in flies over the last 8 years that include roll, pitch, and yaw perturbations. There are crucial experimental differences; you perturb target motion (tracking task) and they provide a mechanical perturbation (i.e. a disturbance), so the experimental topology is different but perhaps their data could give you some insights into the inner-loop control structure?

We appreciate the reviewer’s suggestion and have added a reference to the work of the Cohen group to this the section about flight stabilisation in the Introduction.

5) The paper assumes that is the proboscis mechanoreception has "little or no" feedback contribution. Recent and crucial work by Roth et al. that the authors site quite clearly indicates otherwise for a related species of moth. This manuscript does use data in dim and bright light in the phase before proboscis contact with the nectary for a stationary flower (Figure 2—figure supplement 1), in which case this assumption seems valid, but mechanoreception is known to play a large role in tracking a moving flower (Roth et al). That said, I'm not sure why this rather dubious assumption is needed.

We agree with the reviewer that it the question whether the hawkmoths use mechanosensory feedback from their proboscis to track flowers does not affect our results, and we therefore removed this discussion from our manuscript.

6) Tracking error decreases significantly after reaching to its peak (Figure 3B, Figure 3—figure supplement 1). It is not intuitively clear why or how tracking error would decrease at higher frequencies. Is there any explanation for that? I have some concern that phase lags may have "wrapped" but that the analysis performed didn't identify that wrapping. Could that be possible? See especially Figure 3—figure supplement 1, last row.

Phases are not unwrapped in the analysis and calculation of the tracking error, as these all took place in the complex plane (for detailed discussion of phase wrapping, see Sponberg, 2015, supplementary material. We used exactly the same analysis methods). For the visual presentation in Figure 3—figure supplement 1 we did unwrap the phases – but again, this did not affect any of the tracking error calculations.

Tracking error decreases after the peak, because the animals are barely tracking these frequencies. Since tracking error describes the distance of the moth’s gain and phase from ideal tracking (gain=1, phase=0) in polar coordinates, if the moth has 0 gain, it has by definition a tracking error of 1. This is very nicely visualised in Figure S4 in Sponberg et al., 2015, which I have reproduced for clarification here.

7) As shown in Figure 1D, the power for chirp stimulus is much larger than the power for sum-of- sines. In subsection “Chirp movement”, it is mentioned that "chirp stimulus does not fulfill the linearity criterion".Was this measured quantitatively? If so, how? Maybe if chirp had similar amplitude as the sum-of-sines then it would be linear. Is there a reason to have such large amplitude (and consequently high power) for chirp stimulus?

The linearity required for this type of analysis as laid out by Roth et al., 2014 and Sponberg et al., 2015, demands that the tracking performance (the specific shape of the gain and phase responses) does not depend on the amplitude or phase relationship of the stimulus (within reasonable limits). We therefore determined the linearity of the sum-of-sines stimulus for this hawkmoth species in previous experiments, using a smaller amplitude and a new set of randomised phases for the different flower movement frequencies (see Stöckl et al., 2017, Figure S1).

The chirp stimulus was made to test the limits of the moth’s tracking performance, and forces some (especially in the ablated condition) to abort tracking, as they are not able to follow the movement of the flower at the higher speeds any more. In a sense, it was built to expose non-linearities, in the failure of the moth’s tracking. Very likely, though we did not test this, moths would be able to track higher frequency flower movements at smaller amplitudes, and fewer at larger amplitudes – violating the scaling criterion of linearity. Indeed, it is possible that at smaller stimulus amplitudes, we might find a regime where the chirp stimulus produces responses fulfilling the linearity criterion. However, our stimulus does not, and therefore we chose to use a different type of analysis.

8) The frequency response could be calculated using the data from the chirp stimulus (by using cross spectral density and power spectral density). Also, in Figure 3C, using a second x-axis to show the frequency would be helpful.

A frequency axis in Figure 3C is a very good idea. We added it to the figure.

We also thank the reviewer for this suggestion of using a spectral analysis to look into the tracking of the chirp stimulus in more detail – as we could not use the system identification approach. We have added Figure 3—figure supplement 3 in the supplement, containing the amplitude spectra of the flight tracks of the different antennal conditions, as well as the cross power spectrum density and cross spectrum phase. We added the phase analysis, because the moths in the ablated condition have a rather high jitter in their flight tracks (as also shown in Figure 2). The phase analysis highlights at which frequencies the moth actually had a consistent phase relationship with the flower movement, and shows that moths with ablated flagella did not have it at the higher flower frequencies – in contrast to re-attached and control moths.

In essence, the spectral analysis shows a similar trend as the analysis of the sum-of-sines stimulus, and thus adds detail to our analysis we did not obtain before with just looking at the tracking abortion frequency: the re-attached moths range in between the ablated and the control ones in terms of flower tracking at the higher frequencies. Interestingly, this seemingly small difference in performance between moths with re-attached vs. ablated flagella was enough for most (though not all, see Figure 3D and Figure 3—figure supplement 2) of the re-attached moths to track the entire stimulus to the end like control moths, while all except one of the moths with ablated flagella had to abort flower tracking well before the end of the stimulus. We added the respective discussion to the manuscript as well (subsection “Flagella ablation reduces flower tracking performance at high flower movement frequencies” and “Sum-of-sines movement”).

9) The result for chirp stimulus tracking for control and reattached are very similar (Figure 3C, 3D). What is the explanation for this high similarity?

See our responses to point 2.

10) In the Results section paragraph four, the sentence is vague: "abdominal jitter of moths with re-attached flagella differed significantly from control moths at only one frequency (1.66 Hz)". What's the explanation for that? Why only one frequency? Is this meant to be the same frequency that is listed as 1.7 Hz in subsection “Behavioural experiments”?

We don’t have a mechanistic explanation for why only this particular frequency was statistically significantly different. Since it does not follow an overall trend (all other frequencies were not significantly different), and the frequency sampling was arbitrary (in the sense that we don’t know which frequencies are relevant for hawkmoth flight or if some are more than others), we would not put too much emphasis on this statistical result. However, it is part of our findings and as such we do report it.

The 1.7 Hz is the frequency at which the flower moves, while the 1.66 Hz is the frequency at which the abdomen jitters in the power spectrum. As has been pointed out by reviewer 1, the two different types of frequency analysis are confusing, and we have edited the manuscript to make clearer when we talk about flower frequencies and when we talk about the frequency analysis of the moth’s thorax or abdominal movement in stationary flower trials (see responses reviewer 1, point 1 for details).

11) In paragraph two of subsection “Flagella ablation reduces flower tracking performance at high frequencies in hawkmoths”, based on Figure 3B, the tracking error doesn't look significantly higher? Does it mean statistically significant?

Yes. We added this to all uses of the word “significant” in the manuscript, to avoid confusion (see responses to point 1 for details).

12) Figure 4C, 4D: The plots in three colors have overlapped each other so much that they are unclear. Other methods can be used to show the change of noise in different frequencies.

The fact that the three colours overlap is highlighting our results: there is no significant difference between the three conditions. We agree with the reviewer though that it is hard to identify individual traces, and therefore we have reduced the line width to make identification easier (see Figure 4).

13) In Figure 4C, it is unclear why the dimension of amplitude is shown as Hz.

This is a mistake on our side; the amplitude should be in “mm” and has been corrected.

Reviewer #3:Concerns– A parallel is made by authors between halteres role in Dipteran and flagella in hawkmoth. However, it is surprising that the work of Itai Cohen group was not cited in the Introduction (paragraph two). Cohen was probably the first to measure disturbance rejection in free flying fruitfly around the three axes of rotation (roll, pitch and yaw).

We appreciate the reviewer’s suggestion and have added a reference to the work of the Cohen group to this section of the Introduction.

It is worth noting that halteres have an essential role in the stabilization of the fly, which is not the case in hawkmoth without flagella: the tracking accuracy is degraded but the flight is still stable in hawkmoth. Only the jitter seems to be higher without flagella but the flight remains stable in hover: the tracking seems to be slower indeed. I do not see any instability according to the definition proposed by control engineers. For example in figure 2, hovering flight with ablated flagella is not instable but less accurate. I suggest authors mention a more accurate control than an instable control.

We do not entirely agree with this interpretation. The fact that the jitter in hovering is greater in moths without flagella than in intact moths (as the reviewer points out) is a strong indication that the flight is less stable without antennal mechanosensory feedback – since the moth cannot keep its hovering position reliably and has to use larger corrective movements (quantified as “jitter”). Moreover, when approaching the flower, moths without flagella also fly on much more tortuous paths (Figure 2—figure supplement 1.), giving more indications that flight stability in all flight modes is affected by flagella ablation, not just during flower tracking. We would also argue that the lower accuracy of flower tracking in flagella-ablated moths is due to a lack of control of body position (which also causes the elevated jitter, and the more tortuous flight paths) rather than an impairment in tracking, because all senses required for flower tracking itself are not impaired (vision and potentially proboscis mechanosensation).

We would therefore argue that flight in hawkmoths (in all modes: hovering on one spot, tracking the flower while hovering, forward flight etc.) are less stable (less controlled) without flagella – rather than that tracking is less accurate, as this is only a consequence of the less controlled / more unstable body position. We hope this becomes even clearer with the addition of the control diagram, where we highlight the contributions of the different senses to the different aspects of flight and flower tracking.

– Introduction paragraph four: it is mentioned that antennal mechanosensors play a key role but what kind of role? Authors should add a block diagram to clarify their model and to show clearly the closed-loop control the hawkmoth position:– the tracking error between the insect and target position will be shown– the inner loop based on the antenna block could be shown with respect to the control of the head in body orientation.

We agree with the reviewer that a control diagram is of great benefit to clarify our control hypothesis, and added it to Figure 1. See more detailed discussion on this in the responses to reviewer 2.

As the antennal ablation does not introduce instability but less accurate visual tracking, authors could discuss the fact that antennal mechanoreceptors could act as the prosternal organs in fly to allow the animal to measure the orientation of its head with respect to the insect body (head in boy orientation). As suggested by Viollet and Zeil, 2013, prosternal organs may be involved in a mechanoreceptive feedback on head position relative to the thorax. Antennal mechanoreceptors in hawkmoth could play a similar role: they could just improve the control (accuracy) of the head orientation (gaze).

As explained above, we do not entirely agree with the reviewer’s interpretation of our results that flagella ablation leads to less accurate visual tracking – and suggest that antennal mechanoreception is important for flight control, rather than visual tracking. It is possible, though, that the antenna contribute to head stabilisation – the experiments performed in our study are not suited to shed light on this question. Preliminary data are indeed suggesting that antennae might be involved in head stabilisation, in addition to visual mechanisms. We included the reference in the Discussion.

– Subsection “Sum-of-sines movement”: about the tracking error, please clarify why you used this particular metric defined by equation 1.

Because it is a well-established metric (Roth, Sponberg, and Cowan, 2014; Sponberg et al., 2015; Roth et al., 2016; Stöckl et al., 2017, to describe flight performance in these flower tracking experiments, and allows for a readout of an intuitive metric that combines both the effect on gain and phase.

– What kind of algorithms and software were used to estimate the gain and phase? I recommend having an open access to the programs (code).

We used exactly the same method as specified in the extensive supplementary material in Sponberg et al., 2015, theoretical foundation of which was laid out in Roth, Sponberg and Cowan, 2014. We made this clearer in our Materials and methods section now.

[Editors' note: further revisions were requested prior to acceptance, as described below.]

Essential Revisions– Adding the detailed block diagram figureThe expert reviewers are in strong agreement in consultation that the present control diagram is inaccurate. They recommend that the attached block diagram topology with inner loop be adopted. See the detailed reviews for their reasoning. If the authors insist that a 'simpler' diagram is needed, then the attached block diagram with NO inner loop may be used. Again see the detailed reviews for their reasoning.

We did not intend for our schematic to be a control diagram for flower tracking, which would indeed be inaccurate, as reviewers 2 and 3 point out. We thank reviewer 2 for pointing out that our term “positional perturbations” can be misleading, and therefore agree with the reviewers that it should not remain in its current form. We thank the reviewers for understanding our unease with the detailed control diagram and their proposal of adding a simplified version. We have integrated this control diagram in Figure 1 in the revised version of the manuscript.

– Phase analysis.There are serious concerns about the phase analysis that were not adequately addressed in revision. See the detailed reviews for how these concerns can be addressed.

We have responded to reviewer 2’s concerns about the phase analysis with specific reference to their comments (see below).

Reviewer #2:Block diagram:The authors introduce a different block diagram than what was suggested originally, they mentioned that they don't want to limit their conclusion to flower tracking so they introduce a more general block diagram. […] The left arrow could be something like "Exogenous perturbation" and the return arrow could be "Self-motion feedback" or, maybe "Exafferent perturbation" for the left incoming error and "Reafferent feedback" for the return path.

See our responses to the editor’s point 1.

Phase lag:

*Second, the problem with your currently unwrapped phase is that it is -π/2 at 10Hz, which means that the time lag between the input stimulus and the moth motion is 25ms, which seems extremely fast. I've never seen it in any visuomotor control paper in any species (from external motion to animal motion). If you look for example at Roth et al., 2016 he shows roughly 3π/4 to 2π (almost 360*^o^*) of phase lag, which corresponds to something like 75ms to 100ms which is a lot more sensible. This is very much in line with the feedback delay estimated by Sponberg, 2016 in their "Luminance dependence" paper. It is not possible it is only 25ms of total time lag (delay + low-pass mechanical phase lag). I don't see a problem with the approach but it is a completely unbelievable result. There are many possible sources of this. One problem I've had is when I have data streams from different sources that get temporally offset or that temporally drift. This can introduce leads / lags. Another more likely possibility is that the roll off is so fast that your attempt to unwrap just misses it. See the incredible roll off in Eatai Roth's recent PNAS paper.*

Both the moth’s and the flower’s movement were reconstructed from the same video frames, so it is very unlikely that the observed phase delays were influenced by an offset. We plotted the phase responses without unwrapping the phases, and thus can confirm that unwrapping did not influence the shape of the phase diagram (see below left, the control group at 3000 lux). We further confirmed this by using the same frequency analysis we used for the chirp stimuli in Figure 3–figure supplement 3 (using Matlab’s ‘cspd’ function, which does not unwrap the phases). The example below is also for the control group, 3000 lux, but we have confirmed this for all groups and light conditions. Note that in the analysis to the left, we only extracted gain and phases at the frequencies present in the stimulus – hence the slightly different shapes of the curves.

To avoid similar concerns about the phase unwrapping for the article’s future readers, we have now included the spectral analysis for the sum-of-sines stimulus in the supplement.

**Author response image 2. respfig2:** 

*The fastest latency I've seen estimated is Dyhr et al., 2013 (hawkmoth abdomen) which was 41ms but keep in mind that was just to the abdomen not the entire flight behavior, and it is quite possible that the high-pass "lead" filter helped mask some of that delay. Even still at 10Hz, the phase lag was π (180*^o^*) from stimulus to abdominal movement. The flight mechanics would surely introduce more phase lag.*

We would like to present a few considerations that might render the short phase delays less incredible, compared to those observed previously. An important point to bear in mind is that the previous studies referenced were conducted on *Manduca sexta*, (Roth et al., 2016 and Dyhr et al., 2013), which is a nocturnal hawkmoth, with a much higher body weight and larger size than *M. stellatarum*. Typically, the motor responses of smaller insects are faster than the motor responses of larger insects. Please consider the following:

1) The wing beat frequency of *M. stellatarum* is 80 Hz, which is considerably faster than that of *Manduca sexta* which flaps at 20-25 Hz (see Stoeckl et al., 2017). Thus, the flight mechanics provide much less of a low-pass filter. If flight could be corrected within one wing beat, it would take 40-50 ms in *M. sexta* but only 12 ms in *M. stellatarum*. Moreover, the neural mechanisms controlling flight should be considerably faster if they operate at wing beat frequency.

2) In addition, typically visual responses in diurnal insects are faster than in nocturnal insects. As may be expected, the visual system of the diurnal *M. stellatarum* is distinctly faster than that of the crepuscular/nocturnal *M. sexta* (see Stoeckl et al., 2017).

For these reasons, a response time of 25 ms (roughly two wing beats) may not be as surprising as the reviewer suggests.

Importantly, however, interpreting a specific phase delay as a direct readout for the system’s response delay has several caveats of which we should be aware. Indeed, decrease in phase values might be caused by factors other than the “pure” tracking response of moths.

For instance, a random relation between the moth and the stimulus would lead to an average phase delay of 0 – albeit with a very high variance. This is visible in the analysis of flower tracking with the chirp stimulus in Figure 3—figure supplement 3. Here, one would not argue either that the decrease in phase delay starting around 4Hz is representative of a very fast visual delay – but rather of the moths no longer tracking the stimulus consistently and instead generating random phase delays, which eventually average to 0. A similar effect may occur for the sum-of-sines stimulus. Note especially in the above plot, on the right, how the phase approaches 0 for the “gap” in the stimulus around 7Hz.

Moreover, the moths are not “perfect” in their flight patterns – they always have a certain jitter in their position (not just jitter caused by the wingbeat frequency, but also at frequencies well below it). This is evident in the amplitude spectrum of the thorax’s position in Figure 2. This positional jitter may affect the gain and phase analysis of flower tracking to some degree, as it increases the gain at the jitter frequencies, and corrupts the phase relationship, since the positional jitter is not correlated with flower movement.

A superposition of the phase delays resulting from tracking (which decrease consistently as a function of frequency) with those resulting from positional jitter (which should cluster on average around 0) can result in intermediate phase responses. This may be less of a concern at lower flower movement frequencies, where the amplitude of flower movement far exceeds the positional jitter amplitudes, but it exerts greater influence at higher flower movement frequencies, at which the amplitude of flower movements is only around 0.4 mm.

It is noticeable that the positional jitter is very distinct in the ablated and re-attached groups between 6 and 11 Hz for stationary hovering – and that in these groups the flower tracking gain rises again, starting at 6 Hz, while the phase delay decreases. In the control group, for which the jitter at these frequencies is less pronounced, there is no rise in gain and associated decrease in phase delay, but a levelling in both parameters, which might be caused by the positional jitter. As a note: this levelling in gain and phase responses is something that has been observed in all hawkmoths tested with this paradigm so far (see Sponberg et al. 2015; Roth et al., 2016; Stoeckl et al., 2017).

In our analysis, it is not possible to separate potential positional jitter from “true” flower tracking movements, because we cannot decide which part of the movement is “intentional” and which is not. Hence, it is important to be cautious when interpreting the exact magnitude of the gain and phase delay at the small stimulus amplitudes. In future experiments, we aim to avoid such small flower movement amplitudes, which are confounded with self-generated positional jitter in the animals.

As Reviewer 2 points out, these considerations are important but do not affect the conclusions of our study, especially since the “unusual” parts of the phase response occur at frequencies greater than 5 Hz, at which the tracking gain is very low, and the tracking error in all three species approaches 0. Our analysis of differences in tracking error is focused on lower frequencies, at which the interesting differences between species occur.

Based on the error analysis in the complex plane (which I never really doubted even if I got a bit confused at one point – The explanation re: reduced tracking error at high frequencies is reasonable and I should have realized it before.) I don't think this will affect the main conclusions of the paper but it really does need to be addressed.Reviewer #3:There are just three points I would like to address again:– the new block diagram is not enough accurate. I suggest separating the vision block from the antennal mechanosensors. A visual error can result from a difference between the moth's head orientation and the flower's position. These three signals (head's orientation, flower position and visual error) must be indicated on the diagram. This visual error can then be sent into the vision block, the output of which can be sent to the central integration block. As the mechanosensors seem to play a major role in the stabilisation of the moth, I suggest inserting this block in an inner loop with the motor system block. I agree that this point needs further experiments to determine precisely the function of the mechanosensors.

See our responses to the editor’s point 1.

– Would it be possible that antennae act as a lead compensator (derivator) as the oscillations (jitter) are reduced? This point could be addressed in the discussion.

It is possible that ultimately, the function of the control circuits that the mechanosensory input feeds into, could be described as a lead compensator in control theory. We have not investigated the neural control of the described behaviour and did not build and test a control theory model of the behaviour of nervous system.

– Discussion about the role of the antenna to stabilise the head on the basis of the work of Viollet and Zeil (JEB paper) is not included.

In the previous version of the manuscript, we included a short discussion of the possibility that antennal feedback is required for head stabilisation, and also referred to preliminary data giving evidence for this. We now added the Viollet and Zeil paper to this Discussion (subsection “Flagellar re-attachment improves flight performance”).

[Editors' note: further revisions were requested prior to acceptance, as described below.]

This is an unusual case where the reviewer's rationale for a rather minor required revision requires a rather long argument. Basically this reviewer calls for a caveat to be added to the Discussion in a short paragraph (or a few sentences to an existing paragraph). This caveat will not change the impact of the work, but will help the reader understand the rather remarkable tracking ability of this moth. Revision can be very swift and will not require re-review.Reviewer #2:The authors have done a remarkable job addressing my comments. I particularly appreciate their effort on technical issues such as providing extra data, revised PDFs, etc. [...] The moth seems to be rotating around the thorax. I did see much greater phase lags – on the order of 50ms – when I looked at the flower-to-thorax (near the rear of the thorax) transfer function.

We thank reviewer 2 for their insightful analysis and comments on the moth flower tracking responses. As suggested by the editor, we have added a paragraph of Discussion to the manuscript, were we highlight that the short delays in flower tracking are exceptional when it comes to sensorimotor (and visuomotor) feedback, and that a possible explanation for this might be mechanical coupling between the flower and the hawkmoth via the proboscis. We highlighted the respective paragraph (paragraph two “Antennal mechanosensation and vision operate in different frequency bands”) in colour for easy visibility.